# ROPO: Robust Preference Optimization for Large Language Models

## Abstract

Preference alignment is pivotal for empowering large language models (LLMs) to generate helpful and harmless responses. However, the performance of preference alignment is highly sensitive to the prevalent noise in the preference data. Recent efforts for this problem either marginally alleviate the impact of noise without the ability to actually reduce its presence, or rely on costly teacher LLMs prone to reward misgeneralization. To address these challenges, we propose the **RO**bust **P**reference **O**ptimization (**ROPO**) framework, a novel iterative alignment approach that integrates *noise-tolerance* and *filtering of noisy samples* without the aid of external models. Specifically, ROPO first formulates the training process with adaptive noise reduction as an optimization problem, which can be efficiently solved in an iterative paradigm. Then, to enhance this iterative solving process with noise-tolerance and noise-identification capabilities, we derive a robust loss that suppresses the gradients from samples with high uncertainty. We demonstrate both empirically and theoretically that the derived loss is key to the noise-tolerance and effective filtering of noisy samples. Furthermore, inspired by our derived loss, we propose a robustness-guided rejection sampling technique to compensate for the potential important information in discarded queries. Experiments on three widely-used datasets of dialogue and post-summarization demonstrate that ROPO significantly outperforms existing preference alignment methods in the practical noise setting and under artificial random symmetric noise, with its advantage increasing as the noise rate increases.

## 1 Introduction

Recent research indicates that the significant achievements of Large Language Models (LLMs) in understanding various queries and providing helpful responses [1] rely on the preference alignment, which aligns LLMs' responses with human values and expectations [63; 6; 28]. A typical preference alignment approach is Reinforcement Learning from Human Feedback (RLHF) [7; 76], which first trains a reward model to fit human preferences and subsequently employs an RL algorithm [44] to guide LLMs to generate high-reward responses. However, due to the potential risks of misgeneralized reward modeling [7] and the unstable training [29; 45] of RLHF, various ranking-based methods represented by Direct Preference Optimization (DPO) [42] bypass the explicit reward modeling stage and eschew RL techniques via directly optimizing the implicit reward margins between preferred and dis-preferred responses [69; 59; 47]. Owing to the stable and computationally lightweight supervised learning paradigm, ranking-based methods have emerged as competitive alternatives to RLHF, thus drawing increasing attention recently [45; 63].

Despite their impressive performance on preference alignment, ranking-based methods heavily rely on high-quality preference data, which is costly and limited in practice [21; 8]. First, the noise (e.g., incorrect or ambiguous preferences) in the preference data is unavoidable [58]. Many recent studies have observed the presence of preference noise at levels of 20%-40% across various scenarios [16; 23; 74; 55; 11; 72; 36], whether the annotators are humans or LLMs. Second, the performance of LLMs will significantly deteriorate when trained with noisy preferences [10; 16; 23]. For instance, a 10% increase in the noise rate may lead to a 30% decrease in the performance of DPO in terms of win rate [16]. Therefore, it is highly desirable to develop noise-robust preference alignment techniques.

To address these problems, some recent studies have explored the label smoothing [10; 35] and regularization [16] techniques to alleviate the impact of preference noise. However, these methods can only marginally mitigate the side effects of noise, as the noisy samples are still involved in the training phase. Besides, [16] also attempts to filter out noisy samples but requires another teacher LLM (i.e., a reward model serving as the proxy of the Bradley-Terry model [5]) to assign confidence values to samples, which introduces additional computational costs. Moreover, the teacher LLM may not necessarily provide the correct preference direction on some specific domain [7], and this method is shown to be ineffective at reducing random symmetric noise [16].

In this paper, we propose the **RO**bust **P**reference **O**ptimization (**ROPO**) framework, an iterative alignment approach that unifies *noise-tolerance* and *filtering of noisy samples* without the aid of external models. We first provide a general formulation of learning from noisy preference data as a constrained optimization problem, where we dynamically assign a quality-aware weight for each sample (see Section 3.1). Then, we solve the problem through a provably convergent iterative paradigm, consisting of two alternating steps: *noise-tolerant model training* and *noisy sample filtering*. The main contributions of our method are as follows.

- We propose a robust preference alignment framework that unifies noise-tolerance and filtering of noisy samples. Without the need for any external LLM, the model's robustness and discrimination ability against noisy samples gradually improve as the alternating iterative training proceeds.

- We derive a robust loss function by suppressing the gradients of samples with high uncertainty. The loss contains a noise-aware term, which not only prevents the model from over-fitting to noisy samples but also facilitates identifying noisy samples versus clean samples[1] (see Section 3.2).

- We propose a robustness-guided rejection sampling technique to compensate for the potential important information in discarded queries (see Section 3.3), which improves the data quality and thus leads to further improvement in alignment performance.

- We conduct extensive experiments on three widely-used datasets (i.e., UltraFeedback Binarized, Alpaca Comparison, and TL;DR) with Mistral-7B, Llama-2-7B, Llama-3-8B, Llama-2-13B, and Llama-3-70B. The evaluation results on AlpacaEval, Arena-Hard, and MT-Bench show that the performance of ROPO remains stable in both practical and artificial noisy scenarios.

## 2 PRELIMINARIES AND PROBLEM SETTINGS

Given a query $\mathbf{x} = [x_1, \ldots, x_n]$, an LLM $\pi_\theta$ (with parameters $\theta$) generates a response $\mathbf{y} = [y_1, \ldots, y_m]$, where the tokens $(x_i)_{i=1}^n$ and $(y_j)_{j=1}^m$ come from a predefined vocabulary, in an autoregressive paradigm. Specifically, the model samples $y_j$ from the conditional probability distribution $\pi_\theta(\cdot \mid \mathbf{x}, \mathbf{y}_{1:j-1})$, where $\mathbf{y}_{1:0}$ is null and $\mathbf{y}_{1:j-1} = [y_1, \ldots, y_{j-1}]$ for $j = 2, \ldots, m$. Finally, we can decompose the conditional probability $\pi_\theta(\mathbf{y} \mid \mathbf{x})$ into $\pi_\theta(\mathbf{y} \mid \mathbf{x}) = \prod_{j=1}^m \pi_\theta(y_j \mid \mathbf{x}, \mathbf{y}_{1:j-1})$.

### 2.1 ALIGNMENT OF LARGE LANGUAGE MODELS

Most of the existing LLM alignment frameworks first fine-tune a pre-trained model on high-quality datasets of downstream tasks (e.g., dialogue and post-summarization) via maximum likelihood, in order to teach the model to respond to queries. We denote the supervised fine-tuned model $\pi_{\text{sft}}$. Then, we train the model $\pi_\theta$ (initialized by $\pi_{\text{sft}}$) based on human preference data. Specifically, a preference sample contains a query $\mathbf{x}$, responses $\mathbf{y}_1$ and $\mathbf{y}_2$, and a ranking label $c$ provided by annotators. We use $c = 0$ to indicate that $\mathbf{y}_1$ is preferred to $\mathbf{y}_2$ (denoted $\mathbf{y}_1 \succ \mathbf{y}_2 \mid \mathbf{x}$) and use $c = 1$ to indicate the opposite. We assume that the preference data $(\mathbf{x}, \mathbf{y}_1, \mathbf{y}_2, c)$ is sampled from a distribution $\mathcal{D}$.

A popular formulation of the generation of preferences is the Bradley-Terry (BT) model [5], i.e., $P^*(\mathbf{y}_1 \succ \mathbf{y}_2 \mid \mathbf{x}) = \sigma(r^*(\mathbf{y}_1, \mathbf{x}) - r^*(\mathbf{y}_2, \mathbf{x}))$, where $\sigma$ is the sigmoid function, and $r^*$ is a latent and inaccessible reward function. The key to existing preference learning methods is to explicitly or implicitly approximate $r^*$ or $P^*$. RLHF [37] approximates $r^*$ by training a parameterized reward model $r_\phi$ via maximum likelihood on preference data, then uses the well-trained $r_\phi$ to provide signals for the reinforcement learning of $\pi_\theta$.

---

[1]In Section 3.2, we demonstrate that the cross-entropy loss (i.e., DPO loss) cannot distinguish between noisy samples and clean samples in the context of preference learning, even though it is widely used for learning from noisy data in other scenarios such as image classification [20; 30].

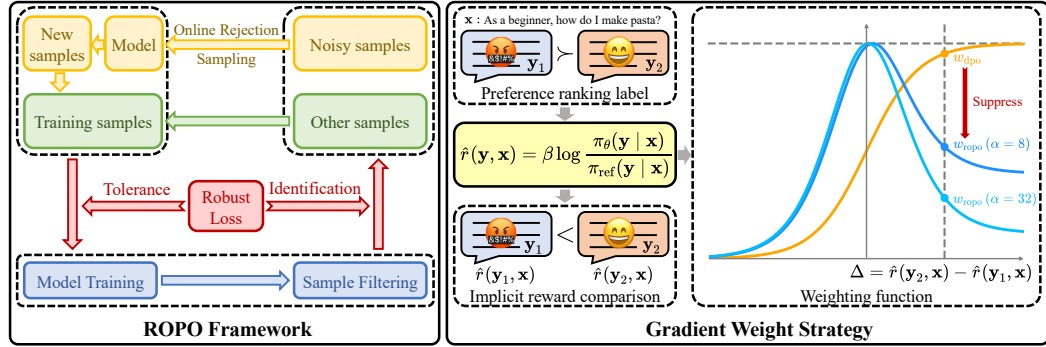

Figure 1: Framework of ROPO and a comparison between the gradient weighting strategies of ROPO and DPO [42]. **Left:** ROPO alternates between noise-tolerant model training and noisy sample filtering and integrates the online rejection sampling paradigm to further improve the data quality. Please see Appendix A for the detailed description and pseudocode of the framework. **Right:** Unlike $w_{\mathrm{dpo}}$, which increases with respect to $\Delta = \hat{r}(\mathbf{y}_2, \mathbf{x}) - \hat{r}(\mathbf{y}_1, \mathbf{x})$, $w_{\mathrm{ropo}}$ decreases when $\Delta$ is large. Given a noisy sample $(\mathbf{x}, \mathbf{y}_1, \mathbf{y}_2, \mathbf{y}_1 \succ \mathbf{y}_2 \mid \mathbf{x})$, whose preference label contradicts the comparison of implicit rewards, ROPO suppresses its gradient. A larger $\alpha$ implies a stronger suppressive effect.

Due to the complexity and instability of RLHF, some recent works [42; 2; 59] directly learn human preferences from offline collected response pairs by optimizing the implicit reward margins between preferred and dis-preferred responses. For example, the objectives of DPO [42] is given by

$$\ell_{\mathrm{dpo}} = -\log \sigma \left( \beta \log \frac{\pi_\theta(\mathbf{y}_1 \mid \mathbf{x})}{\pi_{\mathrm{ref}}(\mathbf{y}_1 \mid \mathbf{x})} - \beta \log \frac{\pi_\theta(\mathbf{y}_2 \mid \mathbf{x})}{\pi_{\mathrm{ref}}(\mathbf{y}_2 \mid \mathbf{x})} \right), \tag{1}$$

where $\mathbf{y}_1 \succ \mathbf{y}_2 \mid \mathbf{x}$, $\beta$ is a hyperparameter, and $\pi_{\mathrm{ref}}$ is a fixed reference model (usually the SFT model). Ranking-based methods are more computationally lightweight and stable than RLHF, thus drawing increasing attention recently. Thus, we mainly focus on ranking-based methods in this paper.

## 2.2 PREFERENCE LEARNING WITH NOISY DATA

Preferences are unavoidably noisy due to the cognitive bias among annotators (see Appendix B for detailed discussion). Thus, we have no access to the clean dataset $D = \{(\mathbf{x}^{(i)}, \mathbf{y}_1^{(i)}, \mathbf{y}_2^{(i)}, c^{(i)})\}_{i=1}^N \sim \mathcal{D}$ and can only obtain a noisy dataset $D_\eta = \{(\mathbf{x}^{(i)}, \mathbf{y}_1^{(i)}, \mathbf{y}_2^{(i)}, \hat{c}^{(i)})\}_{i=1}^N \sim \mathcal{D}_\eta$, where $\hat{c}^{(i)} = c^{(i)}$ with probability $1 - \eta$ and $\hat{c}^{(i)} = 1 - c^{(i)}$ with probability $\eta$.

**Remark.** (1) We assume the random symmetric noise in our **theoretical analysis** because it is the standard assumption for learning from noisy data [32; 70] and existing research on LLM alignment has indicated **the challenges posed by this kind of noise** [16]. Besides, in the context of preference alignment, the symmetric and asymmetric (or class-conditional) noise is equivalent, as the ground truth label is changed if we swap the positions of $\mathbf{y}_1$ and $\mathbf{y}_2$. (2) In addition to this artificially introduced random noise, our experiments also include four types of practical noise settings, covering a variety of unavoidable noises from human and LLM annotations. For more details, please refer to Section 4.1 and Appendix D.3.

## 3 ROBUST PREFERENCE OPTIMIZATION

We propose **RO**bust **P**reference **O**ptimization (**ROPO**), an iterative preference alignment framework. ROPO alternates between *noise-tolerant model training* and *noisy sample filtering*, as shown in Figure 1, which is mathematically equivalent to iteratively solving a constrained optimization problem (Section 3.1). In the model training step, we introduce a robust loss function by suppressing the gradients of samples with high uncertainty, which prevents the model from over-fitting to the noisy preference. In the sample filtering step, we filter out noisy samples based on the magnitude of their training losses. The key to ROPO is that our proposed loss contains a noise-aware term, which not only features noise-tolerance, but also facilitates identifying noisy samples versus clean samples

(Section 3.2). Further, we propose a robustness-guided rejection sampling technique to compensate for the potential important information in discarded queries and thus improve the data quality (Section 3.3). **For detailed proofs of the theorems in this section, please refer to Appendix E.**

### 3.1 A GENERAL FORMULATION

Given $N$ preference samples $\{(\mathbf{x}^{(i)}, \mathbf{y}_1^{(i)}, \mathbf{y}_2^{(i)}, \hat{c}^{(i)})\}_{i=1}^N$, we hope that the weights of noisy samples in the preference optimization are smaller than those of others, thereby reducing the impact of noise on the alignment performance. Without prior knowledge of which samples are noisy, a natural approach would be to assign a dynamic quality-aware weight to each sample and constrain the sum of these weights to a constant, which can also prevent the weights from tending toward zero. Therefore, we formulate learning from noisy preference samples as the following constrained optimization problem:

$$\min_{\theta, \mathbf{w}} \quad \frac{1}{N} \sum_{i=1}^N w_i \ell\left(\theta; \mathbf{x}^{(i)}, \mathbf{y}_1^{(i)}, \mathbf{y}_2^{(i)}, \hat{c}^{(i)}, \pi_\theta\right), \tag{2}$$

$$\text{s.t.} \quad \theta \in \Theta, \quad w_i \in [0,1], \, i = 1, \ldots, N, \quad \sum_{i=1}^N w_i = N_\rho \triangleq \lfloor (1-\rho)N \rfloor,$$

where $w_1, \ldots, w_N$ are dynamic weights, $\Theta$ is compact, and $\rho \in [0,1]$ is the proportion of the samples we aim to filter out. Please note that we minimize Problem (2) with respect to both $\theta$ and $\mathbf{w}$, resulting in a training process that learns the weights adaptively. Hence, we expect that Problem (2) will gradually lead to much smaller weights for noisy samples than those for others. To achieve this, we first analyze the properties of the optimal solution to Problem (2). As shown in Theorem 3.1, Problem (2) admits an optimal solution and the elements in its minimizer $\mathbf{w}^*$ are either 0 or 1.

**Theorem 3.1.** *Assume that $\ell(\theta)$ is continuous on a compact parameter space $\Theta$, then Problem (2) admits an optimal solution $(\theta^*, \mathbf{w}^*)$. Suppose that $\ell\left(\theta^*; \mathbf{x}^{(i_1)}, \mathbf{y}_1^{(i_1)}, \mathbf{y}_2^{(i_1)}, \pi_{\theta^*}\right) < \cdots < \ell\left(\theta^*; \mathbf{x}^{(i_N)}, \mathbf{y}_1^{(i_N)}, \mathbf{y}_2^{(i_N)}, \pi_{\theta^*}\right)$, then $w_{i_k}^* = 1$ for $1 \le k \le N_\rho$ and $w_{i_k}^* = 0$ for $N_\rho < k \le N$.*

We solve Problem (2) in an iterative paradigm, which consists of two alternating steps: model training and sample filtering. In the step of model training, we fix $\mathbf{w}$ and learn model parameters $\theta$. In the step of sample filtering, we fix $\theta$ and assign weights $w_1, \ldots, w_N$ for samples based on their loss values. Because the objective in Problem (2) is non-negative and its value does not increase during the iteration, the iterative solving process is guaranteed to converge.

### 3.2 A NOISE-TOLERANT LOSS

To guarantee the effectiveness of the iterative solving process within the preference alignment framework, we delve into identifying additional conditions that should be imposed on $\ell$. Here, we discuss the properties of $\ell$ in the context of minimizing its expected risks under distributions of noisy and clean preference data, i.e., finding the optimal solutions $\theta^*$ and $\theta_\eta^*$ by solving

$$\theta^* = \arg\min_{\theta \in \Theta} \ \mathbb{E}_{(\mathbf{x}, \mathbf{y}_1, \mathbf{y}_2, c) \sim \mathcal{D}}[\ell(\theta; \mathbf{x}, \mathbf{y}_1, \mathbf{y}_2, c, \pi_\theta)], \tag{3}$$

$$\theta_\eta^* = \arg\min_{\theta \in \Theta} \ \mathbb{E}_{(\mathbf{x}, \mathbf{y}_1, \mathbf{y}_2, \hat{c}) \sim \mathcal{D}_\eta}[\ell(\theta; \mathbf{x}, \mathbf{y}_1, \mathbf{y}_2, \hat{c}, \pi_\theta)]. \tag{4}$$

**Requirement 1: Noise-tolerance.** It cannot be guaranteed that the sample filtering stage will eliminate all noise samples (e.g., when $\rho$ is less than the actual noise proportion in the preference data). Consequently, it is crucial that the presence of noisy preferences does not significantly impact the model training stage, i.e., $\ell$ is noise-tolerant.

**Requirement 2: Distinguishable losses for clean and noisy samples.** As noisy samples generally exhibit larger loss values [30], in the sample filtering step, we filter out the $N - N_\rho$ samples with the largest losses. It is noteworthy that this step takes place midway through training, hence $\ell$ needs to exhibit distinguishable loss values for clean and noisy samples prior to the convergence of the model.

As DPO is one of the most popular preference alignment methods, it is natural for us to explore the effectiveness of the DPO loss $\ell_{\text{dpo}}$ (as shown in Eq. (1)) within our iterative solving process. However, our findings show that $\ell_{\text{dpo}}$ does not satisfy the aforementioned requirements.

**Finding 1: DPO is not noise-tolerant.**

**Theorem 3.2.** *Assume $\eta < \frac{1}{2}$. Consider $\ell_{\text{dpo}}$ and the corresponding minimizer $\theta_\eta^*$ to Problem (4). Given a query $\mathbf{x}$ and responses $(\mathbf{y}_1, \mathbf{y}_2)$, the relationship between the preference probability given by the optimal model, i.e., $P_{\theta_\eta^*}(\mathbf{y}_1 \succ \mathbf{y}_2 \mid \mathbf{x})$, and that given by the BT model, i.e., $P^*(\mathbf{y}_1 \succ \mathbf{y}_2 \mid \mathbf{x})$ is*

$$P_{\theta_\eta^*}(\mathbf{y}_1 \succ \mathbf{y}_2 \mid \mathbf{x}) = P^*(\mathbf{y}_1 \succ \mathbf{y}_2 \mid \mathbf{x}) + (1 - 2P^*(\mathbf{y}_1 \succ \mathbf{y}_2 \mid \mathbf{x})) \cdot \eta, \qquad (5)$$

*hence we have $\left| P_{\theta_\eta^*}(\mathbf{y}_1 \succ \mathbf{y}_2 \mid \mathbf{x}) - P_{\theta^*}(\mathbf{y}_1 \succ \mathbf{y}_2 \mid \mathbf{x}) \right| = 2\eta \left| P^*(\mathbf{y}_1 \succ \mathbf{y}_2 \mid \mathbf{x}) - 1/2 \right|$.*

As shown in Theorem 3.2, the impact of noise on the optimal solution corresponding to $\ell_{\text{dpo}}$ increases as the noise rate increases. Specifically, the difference between the optimal probabilities under noisy and clean distributions, i.e., $\left| P_{\theta_\eta^*}(\mathbf{y}_1 \succ \mathbf{y}_2 \mid \mathbf{x}) - P_{\theta^*}(\mathbf{y}_1 \succ \mathbf{y}_2 \mid \mathbf{x}) \right|$, is proportional to the label flipping probability $\eta$.

**Finding 2: DPO faces challenges in distinguishing between noisy and clean samples.**

**Theorem 3.3.** *Assume $\eta < \frac{1}{2}$. Consider $\ell_{\text{dpo}}$ and the corresponding minimizer $\theta_\eta^*$ to Problem (4). For samples $(\mathbf{x}^{(1)}, \mathbf{y}_1^{(1)}, \mathbf{y}_2^{(1)}, \hat{c}^{(1)} = c^{(1)})$ and $(\mathbf{x}^{(2)}, \mathbf{y}_1^{(2)}, \mathbf{y}_2^{(2)}, \hat{c}^{(2)} = 1 - c^{(2)})$, suppose that $\theta$ is not $\theta_\eta^*$ but satisfies $\max\limits_{i=1,2} \left| P_\theta\left(\mathbf{y}_1^{(i)} \succ \mathbf{y}_2^{(i)} \mid \mathbf{x}^{(i)}\right) - P_{\theta_\eta^*}\left(\mathbf{y}_1^{(i)} \succ \mathbf{y}_2^{(i)} \mid \mathbf{x}^{(i)}\right) \right| < \delta$, then if we want to ensure that $\ell_{\text{dpo}}\left(\mathbf{x}^{(1)}, \mathbf{y}_1^{(1)}, \mathbf{y}_2^{(1)}, \hat{c}^{(1)}\right) < \ell_{\text{dpo}}\left(\mathbf{x}^{(2)}, \mathbf{y}_1^{(2)}, \mathbf{y}_2^{(2)}, \hat{c}^{(2)}\right)$, $\delta$ must satisfy*

$$\delta < \frac{1 - 2\eta}{2}\left(P^*(c^{(1)}) + P^*(c^{(2)}) - 1\right). \qquad (6)$$

As shown in Theorem 3.3, the distance between $\pi_\theta$ and $\pi_{\theta^*}$ we need for $\ell_{\text{dpo}}$ to differentiate between clean and noisy samples decreases as the BT probability approaches 50% and the noise rate increases. Specifically, Eq. (6) shows that the upper bound of $\delta$ is proportional to $(1 - 2\eta)/2$ and $\left(P^*(c^{(1)}) - 1/2 + P^*(c^{(2)}) - 1/2\right)^2$. Due to the intrinsic diversity and stochastic nature of human preferences, the BT distribution is usually not a "hard" distribution with probabilities close to 0 or 1, but rather a "soft" one [52; 51]. This brings difficulties to unconverged DPO-trained model in identifying noisy samples. For example, when $\eta = 30\%$ and $P^*\left(c^{(1)}\right) = P^*\left(c^{(2)}\right) = 60\%$, we need $\delta < 4\%$, which is a challenging requirement for a model that has not yet converged.

**The gradient weighting strategy of DPO may amplify the impact of noise.** Given a sample $(\mathbf{x}, \mathbf{y}_1, \mathbf{y}_2, \hat{c} = 0)$, according to [42], the gradient of $\ell_{\text{dpo}}$ in Eq. (1) is given by

$$\nabla_\theta \ell_{\text{dpo}} = -\beta \underbrace{\sigma\left(\hat{r}(\mathbf{y}_2, \mathbf{x}) - \hat{r}(\mathbf{y}_1, \mathbf{x})\right)}_{w_{\text{dpo}}(\mathbf{x}, \mathbf{y}_1, \mathbf{y}_2)} \cdot \nabla \log \frac{\pi_\theta(\mathbf{y}_1 \mid \mathbf{x})}{\pi_\theta(\mathbf{y}_2 \mid \mathbf{x})}, \qquad (7)$$

where $\hat{r}(\mathbf{y}, \mathbf{x}) = \beta \log \frac{\pi_\theta(\mathbf{y}|\mathbf{x})}{\pi_{\text{ref}}(\mathbf{y}|\mathbf{x})}$ is the implicit reward function of DPO. Intuitively, the greater the discrepancy between the reward function's comparison of $\mathbf{y}_1$ and $\mathbf{y}_2$ and the label $\mathbf{y}_1 \succ \mathbf{y}_2 \mid \mathbf{x}$, the greater the weight $w_{\text{dpo}}(\mathbf{x}, \mathbf{y}_1, \mathbf{y}_2)$ of the DPO gradient becomes. This aggressive weighting strategy can be risky if the label is incorrect, as the model may imply a high uncertainty about the sample by giving a higher reward to $\mathbf{y}_2$ than to $\mathbf{y}_1$, increasing $w_{\text{dpo}}$ and thus amplifying the impact of the noise.

**Conservative gradient weighting strategy.** A simple and straightforward idea is that when the implicit reward margin $\Delta(\mathbf{y}_2, \mathbf{y}_1, \mathbf{x}) \triangleq \hat{r}(\mathbf{y}_2, \mathbf{x}) - \hat{r}(\mathbf{y}_1, \mathbf{x})$ is excessively positive, we should assign a conservative weight to the gradient. Based on this idea, we propose the conservative gradient weight

$$w_{\text{ropo}} = \frac{4\alpha}{(1 + \alpha)^2} \cdot \sigma(\Delta(\mathbf{y}_2, \mathbf{y}_1, \mathbf{x})) \cdot (1 + \alpha\sigma(-\Delta(\mathbf{y}_2, \mathbf{y}_1, \mathbf{x}))), \qquad (8)$$

where $\alpha > 2$ controls the conservatism of weighting and $4\alpha/(1 + \alpha)^2$ is used to normalize the maximum value of $w_{\text{ropo}}$ (see Appendix E.9). As illustrated in Figure 1, unlike the monotonous increase of $w_{\text{dpo}}$, $w_{\text{ropo}}$ decreases when $\Delta(\mathbf{y}_2, \mathbf{y}_1, \mathbf{x})$ is large. Then, the corresponding loss function can be decomposed as

$$\ell_{\text{ropo}} = \int \nabla_\theta \ell_{\text{ropo}} \, d\theta = \frac{4\alpha^2}{(1 + \alpha)^2} \cdot \ell_{\text{na}} + \frac{4\alpha}{(1 + \alpha)^2} \cdot \ell_{\text{dpo}}, \qquad (9)$$

---

[2] As $c^{(1)}$ and $c^{(2)}$ are clean labels, we have $P^*(c^{(1)}) > 1/2$ and $P^*(c^{(2)}) > 1/2$.

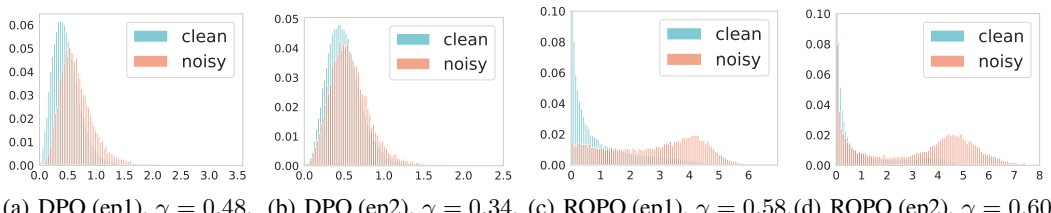

(a) DPO (ep1). $\gamma = 0.48$. (b) DPO (ep2). $\gamma = 0.34$. (c) ROPO (ep1). $\gamma = 0.58$.(d) ROPO (ep2). $\gamma = 0.60$.

Figure 2: Loss distributions of Llama-2-7B trained with DPO and ROPO at different training epochs (ep1 and ep2) on TL;DR. We denote $\gamma$ as the proportion of noisy samples in the 20% of samples that are filtered out. Larger $\gamma$ indicates better discrimination between clean and noisy samples.

where $\ell_{\mathrm{na}} = \sigma \left( \beta \log \frac{\pi_\theta(\mathbf{y}_2|\mathbf{x})}{\pi_{\mathrm{ref}}(\mathbf{y}_2|\mathbf{x})} - \beta \log \frac{\pi_\theta(\mathbf{y}_1|\mathbf{x})}{\pi_{\mathrm{ref}}(\mathbf{y}_1|\mathbf{x})} \right)$ and we omit the constant term of the primitive function (see Appendix E.4 for details). The introduced loss consists of $\ell_{\mathrm{dpo}}$ and a noise-aware term $\ell_{\mathrm{na}}$ whose weight is $\alpha$ times that of $\ell_{\mathrm{dpo}}$. We claim that $\ell_{\mathrm{na}}$ has the following advantages.

**Advantage 1:** $\ell_{\mathrm{na}}$ **is noise-tolerant.**

**Theorem 3.4.** *Assume that $\eta < \frac{1}{2}$. Consider $\ell_{\mathrm{na}}$ and the corresponding minimizer $\theta_\eta^*$ to Problem (4). Given a query $\mathbf{x}$ and responses $(\mathbf{y}_1, \mathbf{y}_2)$, the relationship between the preference probability given by the optimal model, i.e., $P_{\theta_\eta^*}(\mathbf{y}_1 \succ \mathbf{y}_2 \mid \mathbf{x})$, and that given by the BT model, i.e., $P^*(\mathbf{y}_1 \succ \mathbf{y}_2 \mid \mathbf{x})$ is*

$$P_{\theta_\eta^*}(\mathbf{y}_1 \succ \mathbf{y}_2 \mid \mathbf{x}) = \mathbb{I}\left( P^*(\mathbf{y}_1 \succ \mathbf{y}_2 \mid \mathbf{x}) > \frac{1}{2} \right), \tag{10}$$

*hence we have $P_{\theta_\eta^*}(\mathbf{y}_1 \succ \mathbf{y}_2 \mid \mathbf{x}) = P_{\theta^*}(\mathbf{y}_1 \succ \mathbf{y}_2 \mid \mathbf{x})$.*

As shown in Theorem 3.4, contrary to the conclusion in Theorem 3.2 that the optimal solution corresponding to $\ell_{\mathrm{dpo}}$ is affected by the noise, the optimal preference probability corresponding to $\ell_{\mathrm{na}}$, i.e., $P_{\theta_\eta^*}(\mathbf{y}_1 \succ \mathbf{y}_2 \mid \mathbf{x})$, remains unchanged when the label flipping probability $\eta < 1/2$. Specifically, Eq. (10) shows that $P_{\theta_\eta^*}(\mathbf{y}_1 \succ \mathbf{y}_2 \mid \mathbf{x})$ is an indicator function of $P^*(\mathbf{y}_1 \succ \mathbf{y}_2 \mid \mathbf{x}) > 1/2$.

**Advantage 2:** $\ell_{\mathrm{na}}$ **can distinguish noisy samples from clean ones.**

**Theorem 3.5.** *Assume that $\eta < \frac{1}{2}$. Consider $\ell_{\mathrm{na}}$ and the corresponding minimizer $\theta_\eta^*$ to Problem (4). For samples $(\mathbf{x}^{(1)}, \mathbf{y}_1^{(1)}, \mathbf{y}_2^{(1)}, \hat{c}^{(1)} = c^{(1)})$ and $(\mathbf{x}^{(2)}, \mathbf{y}_1^{(2)}, \mathbf{y}_2^{(2)}, \hat{c}^{(2)} = 1 - c^{(2)})$, suppose that $\theta$ is not $\theta_\eta^*$ but satisfies $\max_{i=1,2} \left| P_\theta \left( \mathbf{y}_1^{(i)} \succ \mathbf{y}_2^{(i)} \mid \mathbf{x}^{(i)} \right) - P_{\theta_\eta^*} \left( \mathbf{y}_1^{(i)} \succ \mathbf{y}_2^{(i)} \mid \mathbf{x}^{(i)} \right) \right| < \delta$, then if we want to ensure that $\ell_{\mathrm{na}} \left( \mathbf{x}^{(1)}, \mathbf{y}_1^{(1)}, \mathbf{y}_2^{(1)}, \hat{c}^{(1)} \right) < \ell_{\mathrm{na}} \left( \mathbf{x}^{(2)}, \mathbf{y}_1^{(2)}, \mathbf{y}_2^{(2)}, \hat{c}^{(2)} \right)$, we must have $\delta < \frac{1}{2}$.*

As shown in Theorem 3.5, contrary to the challenging requirement $\ell_{\mathrm{dpo}}$ places on an unconverged model in Theorem 3.3, we can expect that $\ell_{\mathrm{na}}$ yields a larger value for noisy samples than for others as long as the difference between the preference probability given by an unconverged model and that of the optimal model is less than 50%. We verify our theoretical analysis in experiments, as shown in Figure 2. For details about the experiments, please refer to Section 4.2.

**Discussion.** $\ell_{\mathrm{na}}$ is capable of improving noise-tolerance and separating noisy samples from clean samples. However, compared with $\ell_{\mathrm{na}}$, $\ell_{\mathrm{dpo}}$ leads to a "softer" optimal preference probability, which could potentially avoid discrimination against minorities by LLMs. Besides, the aggressive weighting strategy may be useful for clean preference datasets (although they are rare). Thus, it is considered necessary to incorporate a minor component of $\ell_{\mathrm{dpo}}$ into the final loss. From this perspective, the hyperparameter $\alpha$ plays an important role in trading-off between aggressive ($\ell_{\mathrm{dpo}}$) and conservative ($\ell_{\mathrm{na}}$) gradient weighting strategy. Given that the weight of $\ell_{\mathrm{na}}$ is $\alpha > 2$ times greater than that of $\ell_{\mathrm{dpo}}$ (in our experiments and ablations, $\alpha \geq 6$), $\ell_{\mathrm{na}}$ **dominates the optimization process**. Thus, the incorporation of $\ell_{\mathrm{dpo}}$ does not hurt the noise-tolerance and noise filtering too much.

## 3.3 ROBUSTNESS-GUIDED REJECTION SAMPLING

The sample filtering step effectively reduces the proportion of noise but may also discard some important queries. For example, a query designed to eliminate the occupational discrimination in LLMs may be filtered out because the ranking label of its associated responses is incorrect. Thus, inspired by the sample distinguishing ability of our proposed $\ell_{\mathrm{ropo}}$, we propose a rejection sampling technique to compensate for the essential but discarded information and thus improve the robustness of our ROPO framework. Specifically, we sample $K$ responses $\widetilde{\mathbf{y}}_1, \ldots, \widetilde{\mathbf{y}}_K$ to $\mathbf{x}$ for each sample $(\mathbf{x}, \mathbf{y}_1, \mathbf{y}_2)$ that is filtered out and generate $2K$ candidate samples

$$\{(\mathbf{x}, \mathbf{y}_1, \widetilde{\mathbf{y}}_k, \mathbf{y}_1 \succ \widetilde{\mathbf{y}}_k \mid \mathbf{x})\}_{k=1}^K \cup \{(\mathbf{x}, \mathbf{y}_2, \widetilde{\mathbf{y}}_k, \mathbf{y}_2 \succ \widetilde{\mathbf{y}}_k \mid \mathbf{x})\}_{k=1}^K.$$

Then, we compute their loss values and add the sample with the minimum loss to the dataset. Note that we treat the model's responses as dis-preferred ones compared to the original responses, which suppresses the potential unsatisfactory or even harmful information in the model's outputs.

**Discussion.** The rejection sampling is a popular approach of data augmentation to improve the data quality and performance in existing preference alignment methods [12; 31; 65; 67; 60]. Specifically, [12] ranks newly-collected responses based on their rewards and selects the highest ranked one to add to the dataset. To address the issue of the excessively high rejection rate and thus improve the effectiveness of rejection sampling, [65] proposes a multi-step sampling technique, which also requires an external reward model. Besides, [60] and [67] consider rejection sampling for the multi-objective preference alignment, where [60] projects multi-objective reward vectors onto one dimension and then selects samples based on the scalar rewards, while [67] augments samples near the Pareto front of multi-dimensional rewards, leading to a strong multi-objective alignment performance. Compared to the aforementioned methods, which all rely on rewards provided by external models, our robustness-guided rejection sampling technique selects new samples based on loss values that reflect the quality of the samples. Moreover, our technique benefits from being independent of external LLMs, thus leading to computational and memory efficiency.

## 3.4 ROPO FRAMEWORK AND COMPLEXITY ANALYSIS

For the detailed description, pseudocode, and complexity analysis of the overall ROPO framework, please refer to Appendix A due to space limitations in the main text.

## 4 EXPERIMENTS

### 4.1 EXPERIMENTAL SETTINGS

**Tasks and Datasets.** We focus on two dialogue datasets (i.e., UltraFeedback Binarized[3] (UFB) and Alpaca Comparison [39]) and one post-summarization dataset (i.e., Reddit TL;DR [57; 50]). For details about the datasets, please refer to C.1.

**Noise Settings.** As stated in Section 1, the original datasets unavoidably contain noise introduced by annotators (see Appendix D.3 for details about the two related practical noise settings). To further explore the performance of ROPO and baselines under noise, we randomly alter preference labels at different proportions (20% and 40%) within the three datasets to produce more challenging symmetric noise [16]. Besides, in Appendices D.3.1 and D.3.2, we supplement experiments in another two practical settings, where the noise comes from annotators' trust in larger models over smaller models and LLM preference comparisons. Please refer to the supplementary material for more details.

**Baselines, Models, and Hyperparameters.** Our baselines are DPO [42], IPO [2], and two approaches that use the label smoothing technique to alleviate the impact of noise, i.e., rDPO [10] and cDPO [35]. Besides, we supplement experiments on reward modeling in Appendix D.2.

We use Mistral-7B [19] and Llama-2-7B [55] as base models for all baselines and datasets in the main text. For experiments on Llama-2-13B and Llama-3-70B, please refer to Appendix D.1. On UFB, we use Zephyr-7B-SFT-$\beta$ [56] as the SFT model for experiments with Mistral-7B, and adopt the result of Zephyr-7B-$\beta$ [56] on AlpacaEval (90.60) as the performance of DPO under no artificial

---

[3]https://huggingface.co/datasets/HuggingFaceH4/ultrafeedback_binarized

Table 1: Win rates (%) of **different methods vs SFT targets** under different proportions (i.e., 0, 20%, and 40%) of artificial noise, evaluated by GPT-4. The bold font indicates the best result and an underline indicates the second-best result. **Please note that 0% represents no *artificial* noise, which does not mean that the dataset is clean.**

| Dataset | | UFB | | | Alpaca Comparison | | | TL;DR | | |
|---|---|---|---|---|---|---|---|---|---|---|
| Model | Method | 0% | 20% | 40% | 0% | 20% | 40% | 0% | 20% | 40% |
| Mistral-7B | DPO | 90.60 | 86.21 | 82.67 | 73.66 | 70.19 | 65.84 | 63.00 | 56.80 | 49.60 |
| | IPO | 88.45 | 87.32 | 82.86 | 72.92 | 70.81 | 67.33 | 62.00 | 57.00 | 48.80 |
| | rDPO | 88.07 | 87.45 | 84.72 | 72.55 | 72.05 | 70.31 | 62.40 | 58.20 | 52.60 |
| | cDPO | 88.82 | 86.96 | 83.35 | 73.04 | 71.30 | 69.94 | 59.40 | 57.40 | 53.00 |
| | **ROPO** | **91.06** | **88.63** | **87.70** | **75.40** | **76.27** | **74.04** | **79.00** | **77.80** | **75.80** |
| Llama-2-7B | DPO | 68.57 | 66.71 | 62.36 | 53.42 | 50.68 | 48.20 | 56.80 | 42.40 | 35.20 |
| | IPO | 67.70 | 66.09 | 64.35 | 53.54 | 50.56 | 49.19 | 54.20 | 50.80 | 51.60 |
| | rDPO | 68.07 | 67.83 | 65.59 | 52.80 | 51.18 | 50.31 | 54.80 | 54.00 | 50.40 |
| | cDPO | 68.20 | 67.33 | 65.09 | 53.79 | 50.81 | 49.81 | 52.20 | 52.00 | 49.80 |
| | **ROPO** | **68.94** | **69.44** | **66.71** | **55.90** | **54.41** | **54.53** | **78.80** | **78.00** | **79.20** |

noise. In other cases, we fine-tune base models on the preferred responses (SFT targets) to form the SFT models. For details about our baselines, models, and hyperparamters, please refer to Appendix C.2. We run all experiments on 16 NVIDIA A100 GPUs (80 GB).

**Evaluation.** For models trained on UFB and Alpaca Comparison, we evaluate them on the AlpacaEval benchmark [26] by comparing their outputs with those of text-davinci-003 (recommended by the benchmark for comparison). For models trained on TL;DR, we evaluate them by comparing their outputs with the SFT targets (chosen responses) on the test split of TL;DR. Following [42; 56], we employ GPT-4 as the referee for head-to-head comparisons, using the win rate as the metric. The win rate can be computed by $\Omega = \frac{\#(\text{Win})+\#(\text{Tie})/2}{\#(\text{Comparisons})}$, where $\#(\text{Win})$, $\#(\text{Tie})$, and $\#(\text{Comparisons})$ are the numbers of wins, ties, and comparisons, respectively. For evaluation details, experiments on more benchmarks, and human evaluation, please refer to Appendices C.3, D.4, and D.6, respectively.

## 4.2 MAIN RESULTS

**ROPO is robust to noisy preferences.** We present the win rates of different methods vs SFT targets under different proportions of artificial noise in Table 1. From the table, we have several interesting observations: (1) For all preference alignment methods, their win rates show a decreasing trend as the noise rate increases. (2) Compared to the competitors, our proposed ROPO demonstrates a more stable performance under noisy preference data. (3) ROPO consistently outperforms the baselines under different proportions of artificial noise in all the three datasets. Even without artificial noise, ROPO still outperforms DPO by 16.0% on TL;DR and 2.5% on Alpaca Comparison, which indicates that the datasets inherently contain noise. (4) Baselines that use the label smoothing technique (i.e., rDPO and cDPO) mostly outperform other baselines under 20% and 40% artificial noise, but underperform ROPO. We speculate that the reasons for their limited effectiveness are as follows. First, rDPO and cDPO are noise-tolerant only when the hyperparameter $\varepsilon$ exactly equals the proportion of noise and when $\varepsilon = 0.5$, respectively (see Appendix E.7), which is difficult to achieve in practice, as we have no prior knowledge of the exact noise proportion. Second, they do not reduce the presence of noise and thus can only marginally mitigate the side effects of noise. In contrast, ROPO exhibits noise-tolerance without the priors on the noise proportion and iteratively reduces the noise proportion as the training proceeds, thus leading to superior performance to rDPO and cDPO.

**ROPO distinguishes noisy samples from clean samples.** In Section 3.2, we have theoretically shown that $\ell_{\text{na}}$ can distinguish noisy samples from clean ones, while $\ell_{\text{dpo}}$ cannot. Besides, we also claim that the minor incorporation of $\ell_{\text{dpo}}$ in $\ell_{\text{ropo}}$ does not hurt the noise filtering ability. To support our analysis, we report the loss distributions for Llama-2-7B trained with ROPO and DPO on TL;DR in Figure 2. Specifically, for models trained for one (two) epoch, we use the SFT model (the model trained for one epoch) as the reference model and compute the losses for all noisy and clean samples. The results in Figure 2 demonstrate three important observations: (1) $\ell_{\text{ropo}}$ can distinguish between noisy and clean samples by yielding larger values for noisy samples than for others. (2) The

Table 2: Win rates (%) of **ROPO and DPO vs SFT targets** under different proportions (i.e., 0, 20%, and 40%) of artificial noise at different training epochs on TL;DR, evaluated by GPT-4.

| Model | Method | 0% | | | 20% | | | 40% | | |
|-------|--------|-----|-----|-----|-----|-----|-----|-----|-----|-----|
| | | ep1 | ep2 | ep3 | ep1 | ep2 | ep3 | ep1 | ep2 | ep3 |
| Mistral-7B | DPO | 62.60 | 60.20 | 63.00 | 56.80 | 51.00 | 48.60 | 49.60 | 44.40 | 44.60 |
| | ROPO | 75.40 | 75.60 | 79.00 | 68.80 | 76.40 | 77.80 | 61.60 | 70.80 | 75.80 |
| Llama-2-7B | DPO | 49.00 | 53.60 | 56.80 | 42.40 | 38.40 | 39.20 | 32.00 | 35.20 | 33.60 |
| | ROPO | 74.00 | 82.00 | 78.80 | 58.40 | 76.40 | 78.00 | 46.00 | 70.80 | 79.20 |

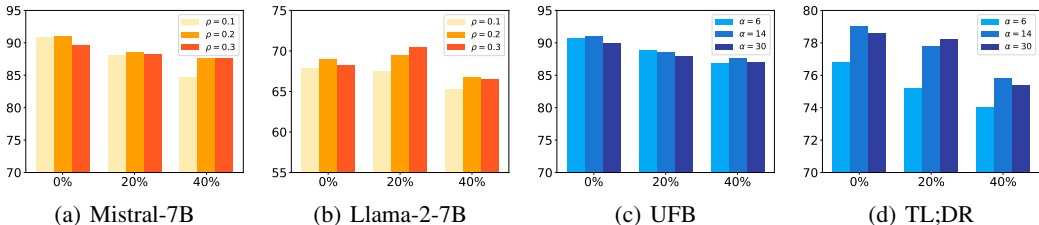

(a) Mistral-7B      (b) Llama-2-7B      (c) UFB      (d) TL;DR

Figure 3: Ablations on $\rho$ and $\alpha$. (a) and (b) respectively show the performance of ROPO-trained Mistral-7B and Llama-2-7B on UFB with different proportions of artificial noise and sample filtering ratio $\rho$. (c) and (d) respectively show the performance of ROPO-trained Mistral-7B on UFB and TL;DR with different proportions of artificial noise and $\alpha$.

distributions of $\ell_{\mathrm{dpo}}$ on noisy and clean samples are similar and the gap between them narrows as training progresses. (3) ROPO has a stronger capability for filtering out noisy samples compared to DPO. Specifically, in the top 20% of samples with the largest $\ell_{\mathrm{ropo}}$, noisy samples make up 60%; whereas in the top 20% of samples with the largest $\ell_{\mathrm{dpo}}$, noisy samples account for about 34%.

**ROPO gradually improves the performance.** In Table 2, we report the win rates of ROPO and DPO vs SFT targets under different proportions of artificial noise at different training epochs on TL;DR. From the results, we find that the performance of ROPO gradually improves as training progresses in most cases, while DPO does not exhibit the same trend. Specifically, the performance of DPO at the second and third epochs is generally lower than that at the first epoch under 20% and 40% artificial noise. As a comparison, the second epoch training of ROPO brings an 8%-24% increase in the win rate, and the third epoch also leads to a 5%-9% improvement under 40% artificial noise. These results demonstrate that the iterative training of ROPO effectively reduces the impact of noise and thus consistently improves the alignment performance.

## 4.3 ABLATIONS

**Effectiveness of components in ROPO.** To evaluate the effectiveness of different components of our ROPO framework, we compare the performance of our proposal with and without: (a) noise-aware term $\ell_{\mathrm{na}}$, (b) noisy sample filtering stage, and (c) rejection sampling stage. As shown in Table 3, all components improve ROPO's performance, validating the rationale of our robust framework. Compared to the aggressive DPO loss, our proposed noise-aware term $\ell_{\mathrm{na}}$ consistently improves the performance, which indicates that a proper trade-off between aggressive and conservative gradient weighting

Table 3: Ablations on different components of ROPO for Mistral-7B on UFB. NSF and RS stand for the noisy sample filtering and rejection sampling stages, respectively.

| Method | 0% | 20% | 40% |
|--------|-----|-----|-----|
| DPO | 90.60 | 86.21 | 82.67 |
| ROPO ($\ell_{\mathrm{na}}$) | 89.19 | 87.58 | 86.34 |
| ROPO ($\ell_{\mathrm{na}}$ + NSF) | 89.44 | 88.20 | **88.07** |
| ROPO ($\ell_{\mathrm{na}}$ + NSF + RS) | **91.06** | **88.63** | 87.70 |

strategy effectively prevents the model from over-fitting to noise. Besides, the results also show that the noisy sample filtering is the most effective part of our method, which also makes ROPO significantly superior to other label smoothing-based methods [10; 35].

**How many noisy samples should we filter out?** The sample filtering ratio $\rho$ is a key factor to the data filtering stage. **In the main experiments, we only report the results with $\rho = 0.2$.** Here, we also present the results of filtering $10\%$ and $30\%$ samples with larger loss values. The results in Figures 3(a) and 3(b) show that better performance could be achieved when filtering $20\%$ or $30\%$ samples. We attribute the reason for this result to the noise ratio in the preference data, which is generally between 20%-30% [16]. There's a substantial risk of eliminating a considerable amount of high-quality data if we set a larger ratio $\rho$. Thus, we recommend using $\rho = 0.2$ in practice.

**Sensitivity to hyperparameters $\alpha$.** The trade-off hyperparameter $\alpha$ controls the importance of the conservative noise-aware term. A larger $\alpha$ indicates a more conservative gradient weighting strategy. As $C \triangleq \lim_{\Delta \to \infty} w_{\text{ropo}}(\Delta) = 4\alpha/(\alpha + 1)^2$, we search the best $\alpha$ in the range of $\{6, 14, 30\}$, which corresponds to $C \in \{1/2, 1/4, 1/8\}$. Then, **we use $\alpha = 14$ in our main experiments** (see Appendix C.2 for the settings of hyperparameters). To explore the effect of $\alpha$, we provide ablations on $\alpha$ in Figures 3(d) and 3(c). As observed, the model's performance remains largely unaffected for $\alpha$ within an appropriate range, as the loss scale does not change significantly (note that $\alpha C \in [2.94, 3.75]$ for $\alpha \in [6, 30]$). Besides, for the dialogue task, a smaller $\alpha$ results in better performance, as a smaller $\alpha$ will lead to more diverse answers. In contrast, a larger $\alpha$ results in better performance in the summarization task. As the summarization task is more objective than the dialogue task, the results are more sensitive to noise, and hence we need a model that is more robust to the noise.

## 5 RELATED WORK

**Preference Alignment of LLMs.** The most representative paradigm of preference alignment is RLHF [76; 37], which involves training a reward model to capture human preferences and then steering LLMs towards producing high-reward responses through RL algorithms [44]. However, in real applications, RL-based methods are complex and prone to instability during training [42; 64; 69]. Therefore, many recent studies have explored more straightforward and stable alternatives for RLHF [69; 42; 47; 61; 28; 27; 63; 71]. Among these studies, the most promising direction is to use a contrastive or ranking loss to calibrate the likelihood of the output sequence. Specifically, RRHF [69] introduces a ranking loss to encourage larger likelihoods for better responses and smaller likelihoods for worse responses. Besides, another important work is DPO [42], which implicitly optimizes the same objective as existing RLHF-based methods and enables human preference alignment directly with a simple cross-entropy loss. In addition to the aforementioned methods using data in the form of $(\mathbf{x}, \mathbf{y}_1, \mathbf{y}_2, c)$, where $c$ is the preference label, some recent studies [13; 15; 8] have also used data in the form of $(\mathbf{x}, \mathbf{y}, \mathbf{c})$, where $c$ is an annotation of the response $\mathbf{y}$, for preference alignment.

**Learning from Noisy Data.** In the era of deep learning, there is an urgent demand for large-scale training samples, and the cost of manually annotating or filtering data is prohibitively expensive in most circumstances [48]. Therefore, learning from noisy data has become increasingly important, which primarily falls into three categories. The first category is sample-selection based methods [53; 41; 38; 49], which identify high-quality samples before training and filter out noisy samples. For example, [53] uses the training dynamics to identify valuable samples. The second category is weighting-based methods, which assign greater weights for important samples and lesser weights for noisy samples [43; 18; 46]. Besides, another important area of research is dedicated to the design of loss functions that are robust to noise [17; 62; 70]. The findings in [17] indicate that the traditional cross-entropy loss is sensitive to the label noise, while symmetric loss functions are robust to such noise. Furthermore, recent advances in LLMs have also underscored the essential role of data quality in both pre-training and supervised fine-tuning (SFT) phases of LLMs [34; 75; 22].

## 6 CONCLUSION

Robust preference optimization is critical for the LLM alignment, as noisy preferences are inevitable in practical scenarios. Unlike existing methods, which rely on label smoothing or external LLMs for the sample selection, we propose a robust preference alignment framework that unifies noise-tolerant model training and effective filtering of noisy samples. Specifically, we incorporate a noise-aware loss term to prevent the model from over-fitting to noise. Besides, we demonstrate that the proposed noise-aware term plays a crucial role in distinguishing noisy samples from clean ones. Furthermore, we propose a robustness-guided rejection sampling technique to compensate for the potential information reduction caused by the filtering stage. We provide extensive theoretical and empirical evidence to demonstrate the effectiveness of our proposed ROPO framework.

## 7 ETHICS STATEMENT

This paper studies LLM alignment with noisy preferences. In our experiments, we introduce artificial noise by randomly flipping the annotated preferences in the datasets and use the obtained noisy datasets to train language models. The models trained on such noisy datasets may tend to output responses that are inconsistent with human values or even harmful. We discourage readers from using models trained on noisy datasets for purposes other than scientific research.

## 8 REPRODUCIBILITY STATEMENT

In this paper, to ensure the reproducibility of our work, we provide key information in the main text and the supplementary material, as summarized as follows.

- **Details about experiments.** We provide details about tasks and datasets, baselines, models, hyperparameters, and the evaluation in Appendix C. We also provide corresponding details for the experiments in Appendix D.
- **Details about the method and theoretical analysis.** In Section 2, we clearly present preliminaries and problem settings. In Section 3, we introduce our method in detail and provide theoretical assumptions and theorems. We also provide rigorous proofs of the claims in Appendix E.

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

# A  ROPO FRAMEWORK

In this section, we describe the overall iterative framework, provide the pseudocode, and analyze the computational cost for ROPO.

---

**Algorithm 1** ROPO

---

**Input:** dataset $D$, $\beta$, $\alpha$, $\rho$, $K$, number of epochs $M$, SFT model $\pi_{\text{sft}}$,
**Initialization:**
   $D^{(0)} \leftarrow D$
   $\pi_\theta^{(0)} \leftarrow \pi_{\text{sft}}$
**for** $m = 1, \ldots, M-1$ **do**
   $\pi_{\text{ref}}^{(m)} \leftarrow \pi_\theta^{(m-1)}$ with frozen parameters $\theta$
   ▶ **Noise-tolerant Training:**
      Obtain $\pi_\theta^{(m)}$ by training $\pi_\theta^{(m-1)}$ on $D^{(m-1)}$ with $\pi_{\text{ref}}^{(m)}$ and $\ell_{\text{ropo}}$ in Eq. (9) for one epoch
   ▶ **Noisy Sample Filtering:**
      Compute $\ell_{\text{ropo}}$ with $\pi_\theta^{(m)}$ and $\pi_{\text{ref}}^{(m)}$ for $D$
      $D_{\text{top}-\rho}^{(m)} \leftarrow$ samples with top-$\rho$ ROPO loss value in $D$
      $D_{\text{bot}-(1-\rho)}^{(m)} \leftarrow$ samples with bottom-$(1-\rho)$ ROPO loss value in $D$
   ▶ **Robustness-guided Rejection Sampling:**
      $D_{\text{new}} \leftarrow \varnothing$
      **for** $(\mathbf{x}, \mathbf{y}_1, \mathbf{y}_2)$ in $D_{\text{top}-\rho}^{(m)}$ **do**
         Sample responses $\widetilde{\mathbf{y}}_1, \ldots, \widetilde{\mathbf{y}}_K$ to $\mathbf{x}$ using $\pi_\theta^{(m)}$
         $D_{\text{cand}} \leftarrow \{(\mathbf{x}, \mathbf{y}_1, \widetilde{\mathbf{y}}_k, \mathbf{y}_1 \succ \widetilde{\mathbf{y}}_k \mid \mathbf{x})\}_{k=1}^K \cup \{(\mathbf{x}, \mathbf{y}_2, \widetilde{\mathbf{y}}_k, \mathbf{y}_2 \succ \widetilde{\mathbf{y}}_k \mid \mathbf{x})\}_{k=1}^K$
         $D_{\text{new}} \leftarrow D_{\text{new}} \cup \{\arg\min_{\mathbf{z} \in D_{\text{cand}}} \ell_{\text{ropo}}(\mathbf{z}, \pi_\theta^{(m)})\}$
      **end for**
      $D^{(m)} \leftarrow D_{\text{bot}-(1-\rho)}^{(m)} \cup D_{\text{new}}$
**end for**
$\pi_{\text{ref}}^{(M)} \leftarrow \pi_\theta^{(M-1)}$ with frozen parameters $\theta$
Obtain $\pi_\theta^{(M)}$ by training $\pi_\theta^{(M-1)}$ on $D^{(M-1)}$ with $\pi_{\text{ref}}^{(M)}$ and $\ell_{\text{ropo}}$ in Eq. (9) for one epoch
**Output:** $\pi_\theta^{(M)}$

---

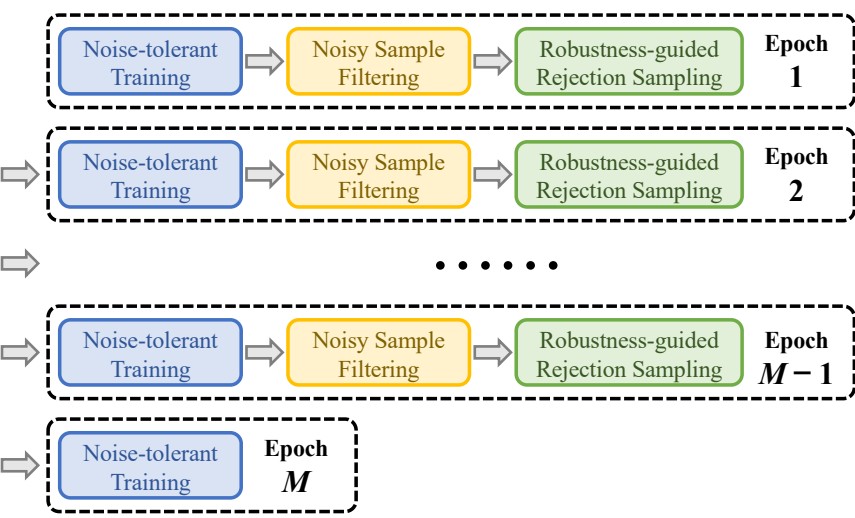

Figure 4: The iterative process of ROPO.

As shown in Figure 4 and Algorithm 1, ROPO iteratively carries out three stages: noise-tolerant training, noisy sample filtering, and robustness-guided rejection sampling. Specifically, in the 1st to

Table 4: Notations in the computational cost analysis.

| Notation | Description |
|---|---|
| $N, M, \rho, K$ | Please see Algorithm 1 |
| $C_{\text{ROPO}}$ | Cost of ROPO |
| $C_{\text{non-it}}$ | Cost of non-iterative methods |
| $C_{\text{tr}}$ | Cost of (noise-tolerant) training per epoch |
| $C_{\text{fil}}$ | Cost of noisy sample filtering per epoch |
| $C_{\text{rs}}$ | Cost of robustness-guided rejection sampling per epoch |
| $C_{\text{loss}}$ | Cost of computing the loss for a sample $(\mathbf{x}, \mathbf{y}_1, \mathbf{y}_2)$ without gradient propagation |
| $C_{\text{gen}}$ | Cost of generating a response $\mathbf{y}$ for a query $\mathbf{x}$ |
| $C_{\text{forward}}$ | Cost of computing the log-likelihood for a query-response pair $(\mathbf{x}, \mathbf{y})$ |
| $C_{\text{backward}}$ | Cost of computing the gradient and updating parameters for a sample $(\mathbf{x}, \mathbf{y}_1, \mathbf{y}_2)$ |

$(M-1)$th epochs, we first train the model using the ROPO loss $\ell_{\text{ropo}}$. After training for one epoch, we compute the value of ROPO loss for all samples in the original dataset and divide them into two subsets (i.e., $D_{\text{top}-\rho}$ and $D_{\text{bot}-(1-\rho)}$) according to their loss values. Then, the robustness-guided rejection sampling stage generates new samples $D_{\text{new}}$ based on $D_{\text{top}-\rho}$. The new samples are used together with $D_{\text{bot}-(1-\rho)}$ as training samples for the next epoch. In the last epoch, we only perform the noise-tolerant training stage and then get the final model.

**Computational Cost Analysis.** ROPO introduces additional costs for the noisy sample filtering and robustness-guided rejection sampling stages compared with non-iterative methods. However, these additional costs are acceptable compared to the training cost and almost negligible in the entire chain of real-world large-scale LLM training. The notations in the computational cost analysis is shown in Table 4. We have

$$\frac{C_{\text{ROPO}}}{C_{\text{non-it}}} = \frac{MC_{\text{tr}} + (M-1)(C_{\text{fil}} + C_{\text{rs}})}{MC_{\text{tr}}}. \tag{11}$$

Since the main cost of the noisy sample filtering stage per epoch is to compute the loss of $N$ samples, we have $C_{\text{fil}} \approx NC_{\text{loss}} \approx 4NC_{\text{forward}}$. As for the rejection sampling stage, the main costs per epoch come from $\rho NK$ response generation and $2\rho NK$ loss computations, hence $C_{\text{rs}} \approx \rho NKC_{\text{gen}} + 2\rho NKC_{\text{loss}} \approx \rho NKC_{\text{forward}} + 8\rho NKC_{\text{forward}} = 9\rho NKC_{\text{forward}}$. Because the training process mainly involves loss computation for two models (i.e., the reference model and the model being trained) and gradient propagation, we have $C_{\text{tr}} \approx N(C_{\text{loss}} + C_{\text{backward}}) \approx N(4C_{\text{forward}} + C_{\text{backward}})$. Therefore, Eq. (11) leads to

$$\frac{C_{\text{ROPO}}}{C_{\text{non-it}}} \approx \frac{M(4C_{\text{forward}} + C_{\text{backward}}) + (M-1)(4 + 9\rho K)C_{\text{forward}}}{M(4C_{\text{forward}} + C_{\text{backward}})}$$

$$= 1 + \frac{(4 + 9\rho K)(M-1)}{M} \cdot \frac{C_{\text{forward}}}{4C_{\text{forward}} + C_{\text{backward}}}$$

$$= 1 + \frac{(4 + 9\rho K)(M-1)}{M} \cdot \frac{1}{4 + C_{\text{backward}}/C_{\text{forward}}}, \tag{12}$$

where the ratio $C_{\text{backward}}/C_{\text{forward}}$ is approximately $2-3$ for LLMs. Take $\rho = 0.2, K = 2, M = 3$ as an example, without considering inference acceleration, we can estimate that $C_{\text{ROPO}} \approx 1.6C_{\text{non-it}}$. In practice, we can use inference acceleration methods to increase $C_{\text{backward}}/C_{\text{forward}}$, thereby further reducing the additional cost of ROPO. Compared with the computational cost of the entire chain of real-world LLM training (including continual pre-training and SFT), the additional cost is almost negligible.

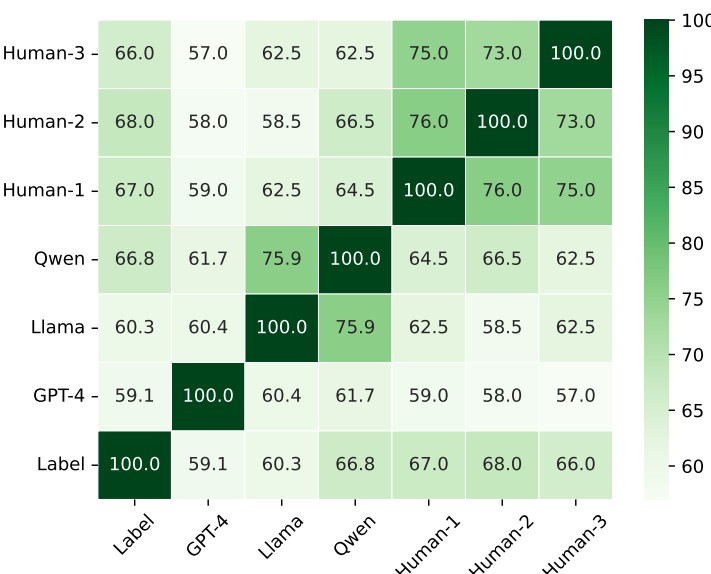

Figure 5: The inter-annotator agreement heap map on the TL;DR dataset. The "Label", "Llama", and "Qwen" refer to the original labels in the dataset, Llama-2-70B, and Qwen-Max, respectively. We assess the preferences of human annotators across 200 randomly selected samples and extend the evaluation to 1,000 samples for LLMs, which include the initial 200 samples.

Table 5: The estimated noise rate in commonly-seen datasets. This table is from [16].

| Dataset | Noise rate (%) | Reference |
|---------|----------------|-----------|
| MT-Bench | 15.0-37.0 | [73] |
| TL;DR | 21.3-27.0 | [23] |
| CBArena | 22.0-36.0 | [73] |
| AntHH | 27.9-30.9 | [23] |
| SHP | 35.5-41.9 | [11] |
| WebGPT | 34.8 | [11] |

## B  DISCUSSION ON PREFERENCE NOISE

Due to the inherent differences in annotators' preferences, the preference noise is usually unavoidable. In this section, we discuss the definition and identification of preference noise.

Before giving the definition of preference noise, we invite our readers to pay attention to the following two points.

1. This paper focuses on **noisy preferences** rather than the more general noisy preference data. The former refers specifically to the noise in preference labels, while the noise corresponding to the latter may come from multiple factors such as preference labels, text quality, and the matching degree between queries and responses. It is interesting and meaningful to study a wider range of noisy data, but it is beyond the scope of our paper and related work [35; 10; 16].

2. Like related work [35; 10; 16], this paper is based on the **Bradley-Terry (BT) model**. The BT model assumes the existence of a "gold", latent, and inaccessible reward model $r^*$. Then, we can express the BT preference probability $P^*(y_1 \succ y_2 \mid x)$ for a sample $(x, y_1, y_2)$ using the reward model $r^*$. Intuitively, the BT model assumes that there are mainstream preferences in human society that reflect values such as peace, friendliness, honesty, etc. Differently, there are also studies on multifaceted or multidimensional preferences [33; 24], but defining noise for them is challenging because it is difficult to have a "ground truth" label. Therefore, our following discussion is based on the assumption of the BT model.

**Definition of preference noise.** For a sample $(\mathbf{x}, \mathbf{y}_1, \mathbf{y}_2, \hat{c})$, if $P^*(\hat{c}) > 0.5$, then the sample is clean; otherwise, the sample is noisy. Because the BT model usually represents preferences that are consistent with mainstream values of human society, so the formation of such noise is usually caused by the personal preferences or cognitive biases of annotators. Please note that the annotators can be humans or LLMs.

**Identification and detection of preference noise.** As mentioned above, the definition of preference noise is based on the inaccessible reward model, so we can never identify preference noise accurately. However, we can estimate the noise rate by *using advanced LLMs as the proxy for the BT model* or *computing the inter-annotator agreement*.

- Using advanced LLMs as the proxy for the BT model [16]. Given a dataset, we can prompt advanced LLMs (e.g., GPT-4, Llama-3-70B-Instruct [14], and Qwen-2-72B-Instruct [66]) to identify the noise. For example, we can provide them with rules and ask them to rate or rank the responses in the dataset. If a sample's new label is different from its original label, it is identified as noisy. The stronger the proxy LLM, the more reliable the noise identification.

- Computing the inter-annotator agreement [37; 58; 4]. We can employ different annotators (humans or LLMs) to relabel the dataset and calculate the agreement between them. For this approach, we should try to ensure that all annotators have the same criteria, and similar cognition and ability. Suppose that we have $n$ annotators and the agreement between annotators $i$ and $j$ is $0 \leq a_{ij} \leq 1$, then the estimated noise rate is $\frac{1}{n(n-1)} \sum_{i,j=1, i \neq j}^{n}(1 - a_{ij})$. Take the TL;DR dataset as an example. We employ GPT-4, Llama-2-70B [55], Qwen-Max [3], and three human annotators to relabel the TL;DR dataset. The human annotators are three of the four volunteers mentioned in Appendix D.6. The inter-annotator agreement heat map is shown in Figure 5, which indicates an estimated noise rate of 17.6%.

Besides, Table 5 from [16] summarizes the estimated noise rate in some commonly-seen datasets. As can be seen, the existence of preference noise is ubiquitous and cannot be ignored, which highlights the importance of studying robust preference optimization approaches.

## C  MORE DETAILS ABOUT EXPERIMENTS

### C.1  TASKS AND DATASETS

We run experiments on two dialogue datasets (i.e., UltraFeedback Binarized and Alpaca Comparison) and one post summarization dataset (i.e., TL;DR).

- The UltraFeedback Binarized dataset[4] is a pre-processed version of the UltraFeedback dataset [11], which contains 64,000 prompts and each prompt has four model responses from various LLMs. Based on the score assigned by GPT-4, [56] selects two responses for each prompt and construct UltraFeedback Binarized for the preference alignment.

- The Alpaca Comparison dataset contains 52,000 queries from the widely-used Stanford Alpaca dataset [54]. [39] generates several responses using GPT-4 and other LLMs including text-davinci-003 to each query and employs GPT-4 to assign a score for each response.

- In the TL;DR dataset, each prompt is a forum from Reddit, and the model is required to summarize the given forum. Following [42], we use the Reddit TL;DR summarization dataset [57] along with human preferences collected by [50].

### C.2  BASELINES, MODELS, AND HYPERPARAMETERS

**Baselines.** Our baselines are DPO [42], IPO [2], and two approaches that use the label smoothing technique to alleviate the impact of noise, i.e., rDPO [10] and cDPO [35].

---

[4]https://huggingface.co/datasets/HuggingFaceH4/ultrafeedback_binarized

Specifically, given a preference data $(\mathbf{x}, \mathbf{y}_1, \mathbf{y}_2)$ with the ranking label $\mathbf{y}_1 \succ \mathbf{y}_2 \mid \mathbf{x}$, the objectives of our baselines are

$$\ell_{\text{dpo}} = -\log \sigma\left(\beta \log \frac{\pi_\theta(\mathbf{y}_1 \mid \mathbf{x})}{\pi_{\text{ref}}(\mathbf{y}_1 \mid \mathbf{x})} - \beta \log \frac{\pi_\theta(\mathbf{y}_2 \mid \mathbf{x})}{\pi_{\text{ref}}(\mathbf{y}_2 \mid \mathbf{x})}\right), \tag{13}$$

$$\ell_{\text{ipo}} = \left(\log \frac{\pi_\theta(\mathbf{y}_1 \mid \mathbf{x})}{\pi_{\text{ref}}(\mathbf{y}_1 \mid \mathbf{x})} - \log \frac{\pi_\theta(\mathbf{y}_2 \mid \mathbf{x})}{\pi_{\text{ref}}(\mathbf{y}_2 \mid \mathbf{x})} - \frac{1}{2\beta}\right)^2, \tag{14}$$

$$\ell_{\text{rdpo}} = -\frac{1-\varepsilon}{1-2\varepsilon}\log \sigma\left(\beta \log \frac{\pi_\theta(\mathbf{y}_1 \mid \mathbf{x})}{\pi_{\text{ref}}(\mathbf{y}_1 \mid \mathbf{x})} - \beta \log \frac{\pi_\theta(\mathbf{y}_2 \mid \mathbf{x})}{\pi_{\text{ref}}(\mathbf{y}_2 \mid \mathbf{x})}\right)$$
$$+ \frac{\varepsilon}{1-2\varepsilon}\log \sigma\left(\beta \log \frac{\pi_\theta(\mathbf{y}_2 \mid \mathbf{x})}{\pi_{\text{ref}}(\mathbf{y}_2 \mid \mathbf{x})} - \beta \log \frac{\pi_\theta(\mathbf{y}_1 \mid \mathbf{x})}{\pi_{\text{ref}}(\mathbf{y}_1 \mid \mathbf{x})}\right), \tag{15}$$

$$\ell_{\text{cdpo}} = -(1-\varepsilon)\log \sigma\left(\beta \log \frac{\pi_\theta(\mathbf{y}_1 \mid \mathbf{x})}{\pi_{\text{ref}}(\mathbf{y}_1 \mid \mathbf{x})} - \beta \log \frac{\pi_\theta(\mathbf{y}_2 \mid \mathbf{x})}{\pi_{\text{ref}}(\mathbf{y}_2 \mid \mathbf{x})}\right)$$
$$- \varepsilon \log \sigma\left(\beta \log \frac{\pi_\theta(\mathbf{y}_2 \mid \mathbf{x})}{\pi_{\text{ref}}(\mathbf{y}_2 \mid \mathbf{x})} - \beta \log \frac{\pi_\theta(\mathbf{y}_1 \mid \mathbf{x})}{\pi_{\text{ref}}(\mathbf{y}_1 \mid \mathbf{x})}\right), \tag{16}$$

where $\varepsilon \in (0, \frac{1}{2})$ and $\beta \in (0, 1)$ are hyperparameters.

**Models.** We use Mistral-7B [19] and Llama-2-7B [55] as base models for all baselines and datasets. On UFB, we use Zephyr-7B-SFT-$\beta$ [56] as the SFT model for experiments with Mistral-7B, and adopt the result of Zephyr-7B-$\beta$ [56] on AlpacaEval (90.60) as the performance of DPO under no artificial noise. In other cases, we fine-tune base models on the preferred responses (SFT targets) to form the SFT models.

**Hyperparameters.** We run all experiments on 16 NVIDIA A100 GPUs (80 GB). Unless otherwise noted, we use a global batch size of 512 to train all models. For all hyperparameters **except for $\varepsilon$ of label smoothing**, **we search for the best one on each dataset without artificial noise and use the same setting for 20% and 40% artificial noise**.

For all methods, we search the best learning rate in {1e-5, 5e-6, 1e-6, 5e-7, 1e-7} and the best $\beta$ in {0.1, 0.5}. We find that the best performing learning rate is 1e-6, and the best $\beta$ for dialogue and post summarization are 0.1 and 0.5, respectively. This conclusion is consistent with that in [42].

**For ROPO, we use $\alpha = 14$ and $\rho = 0.2$ in the main experiments.** In ablations (Section 4.3), we tune $\alpha$ in {6, 14, 30}, which makes $\frac{4\alpha}{(1+\alpha)^2}$ be around $\frac{1}{2}, \frac{1}{4}, \frac{1}{8}$, respectively, and tune $\rho$ in {0.1, 0.2, 0.3}. We set $K = 3$ for the rejection sampling. For rDPO and cDPO, we search the best $\varepsilon$ in {0,05, 0.1, 0.15, 0.2, 0.25, 0.3, 0.35, 0.4, 0.45} for each dataset and each proportion of artificial noise.

## C.3 EVALUATION

For models trained on UFB and Alpaca Comparison, we evaluate them on the AlpacaEval benchmark [26]—a widely used dialogue benchmark—by comparing their outputs with those of text-davinci-003 (recommended by the benchmark for comparison). AlpacaEval contains 805 queries in various domains and exhibit a strong concordance with ground truth human annotators. For TL;DR, we randomly select 500 queries from the test split of it and evaluate ROPO and baselines by comparing their outputs with the chosen responses (SFT targets) for the queries.

Following existing studies [42; 56], we employ GPT-4 as the referee to conduct head-to-head comparisons, using the win rate as the metric. On AlpacaEval, we conduct evaluations using the API provided by AlpacaEval. On TL;DR, we use the following prompt, which is similar to that used by AlpacaEval, to conduct GPT-4 evaluation.

Table 6: Win rates (%) of **different methods vs SFT targets** under different proportions (i.e., 0 and 20%) of artificial noise, evaluated by GPT-4 on AlpacaEval.

| Dataset | | UFB | |
| --- | --- | --- | --- |
| Model | Method | 0% | 20% |
| | DPO | 82.98 | 80.50 |
| | IPO | 81.99 | 79.75 |
| Llama-2-13B | rDPO | 81.37 | 80.87 |
| | cDPO | 82.36 | 80.50 |
| | **ROPO** | **83.23** | **82.98** |

Table 7: Win rates (%) of **ROPO/DPO vs SFT targets** under different proportions (i.e., 0 and 20%) of artificial noise, evaluated by GPT-4 on AlpacaEval.

| Dataset | | UFB | |
| --- | --- | --- | --- |
| Model | Method | 0% | 20% |
| Llama-2-70B | DPO | 94.29 | 88.70 |
| | **ROPO** | **95.53** | **94.04** |

```
You are a helpful assistant that ranks models by the
quality of their summaries of given forum posts.

I want you to create a leaderboard of different of
large-language models.  To do so, I will give you the
instructions (forum posts) given to the models, and the
responses of two models.  Please rank the models based on
which responses would be preferred by humans.

Here is the post:
<Forum Post>

Here are the outputs of the models:
Model 1:  <Summary 1>
Model 2:  <Summary 2>

Now please rank the models by the quality of their answers,
so that the model with rank 1 has the best output.  Please
provide the ranking that the majority of humans would give.
Your response should use the format:
Better:  <Model 1 or Model 2>
```

# D MORE EXPERIMENTS

## D.1 EXPERIMENTS ON LLAMA-2-13B-BASE AND LLAMA-3-70B-BASE

To evaluate ROPO and baselines on models larger than 7B, we supplement experiments on Llama-2-13B-Base and Llama-3-70B-Base.

**Experiments on Llama-2-13B-Base.** We run SFT on UltraChat-200k for one epoch with the learning rate of 1e-5, the global batch size of 128, the weight decay of 0.1, and a cosine-type learning rate scheduler. Then, we fine-tune the SFT model with ROPO and baselines for two epochs on UFB (under artificial noise ratio of 0 and 20%) with the learning rate of 1e-6 and the global batch size of 512. In the experiments, we fix $\alpha = 14$ and $\rho = 0.2$ for ROPO without tuning them, and tune $\beta$ in $[0.1, 0.5, 1.0]$ for IPO and tune $\varepsilon$ in $[0.1, 0.2, 0.3, 0.4]$ for cDPO and rDPO. The results are shown in Table 6.

Table 8: Win rates (%) of **different methods vs SFT targets** under different proportions (i.e., 0 and 20%) of artificial noise, evaluated by GPT-4 on AlpacaEval.

| Dataset | | UFB | |
|---|---|---|---|
| Model | Method | 0% | 20% |
| Mistral-7B | DPO-RM | 69.69 | 68.32 |
| | cPPO-RM | 68.45 | 67.95 |
| | rPPO-RM | 68.70 | 67.33 |
| | **ROPO-RM** | **69.94** | **70.43** |

Table 9: Win rates (%) of **different methods vs SFT targets** under noise coming from the annotators' trust in larger models over smaller ones, evaluated by GPT-4 on AlpacaEval.

| Dataset | | UFB |
|---|---|---|
| Model | Method | |
| Mistral-7B | DPO | 75.16 |
| | IPO | 72.55 |
| | cDPO | 76.27 |
| | rDPO | 78.26 |
| | **ROPO** | **80.50** |

**Experiments on Llama-3-70B-Base.** We run SFT on UltraChat-200k for one epoch with the learning rate of 1e-5, the global batch size of 128, the weight decay of 0.1, and a cosine-type learning rate scheduler. Then, we fine-tune the SFT model with ROPO and DPO for two epochs on UFB (under artificial noise ratio of 0 and 20%) with the learning rate of 5e-7 and the global batch size of 512. We fix $\alpha = 14$ and $\rho = 0.2$ for ROPO without tuning them. The results are shown in Table 7. From the results we can conclude that: (1) 70B models outperform 7B/13B models in terms of win rate. However, the performance of the models trained with DPO still has a non-negligible drop under 20% artificial noise. (2) Our ROPO still significantly exceeds DPO on the scale of 70B.

### D.2 EXPERIMENTS ON REWARD MODELING

In the main text of our paper, the baselines are reward-free. Considering the reward modeling (RM) still plays an important role in many real-world LLM applications, although RM is not our focus, we supplement experiments on RM with Mistral-7B-Base to test the potential of ROPO in scenarios including reward modeling. Given a sample $(\mathbf{x}, \mathbf{y}_1, \mathbf{y}_2, c = 0)$, if we denote $P = \sigma(r(\mathbf{x}, \mathbf{y}_1) - r(\mathbf{x}, \mathbf{y}_2))$, then the RM-training losses of ROPO and our baselines are as follows.

- DPO-RM: $-\log P$
- cPPO-RM: $-(1 - \varepsilon)\log P - \varepsilon \log(1 - P)$
- rPPO-RM: $-\frac{1-\varepsilon}{1-2\varepsilon}\log P + \frac{\varepsilon}{1-2\varepsilon}\log(1 - P)$
- ROPO-RM (Ours): $-(4\alpha/(1+\alpha)^2) \cdot P + (4\alpha^2/(1+\alpha)^2) \cdot (1 - P)$

We train Mistral-7B-v0.1 on UFB for two epochs with the aforementioned losses to obtain reward models. Then, we use Best of $N$ Sampling ($N = 16$) to generate responses based on RMs and Mistral-7B-SFT-Beta (SFT model). We use the learning rate of 5e-6, the batch size of 512, and a cosine-type learning rate scheduler. The results are shown in Table 8

### D.3 MORE PRACTICAL NOISE SETTINGS

The experiments in the main text cover two types of practical noise as follows.

1. **Practical noise coming from human comparisons.** In the original TL;DR dataset, the preferences are labeled by human annotators who compare the post-summaries generated by different models in pairs. This leads to unavoidable noise due to the diversity of human preferences.

2. **Practical noise coming from LLM (GPT-4) rating.** Each query (instruction) in the original UltraFeedback dataset has four responses coming from different models. GPT-4 scores them

Table 10: Win rates (%) of **different methods vs SFT targets** under noise coming from LLM preference comparisons, evaluated by GPT-4 on AlpacaEval.

| Dataset | | UFB |
|---|---|---|
| Model | Method | |
| Mistral-7B | DPO | 84.22 |
| | IPO | 84.84 |
| | cDPO | 85.22 |
| | rDPO | 86.21 |
| | **ROPO** | **88.07** |

based on criteria such as instruction-following, honesty, helpfulness, etc. Then, for each query, the highest ranked response is selected as "preferred", and one of the remaining responses is randomly selected as "dis-preferred". This leads to unavoidable noise due to the bias of GPT-4.

In this section, we explore another two practical noise settings in Appendices D.3.1 and D.3.2.

### D.3.1 EXPERIMENTS UNDER NOISE COMING FROM ANNOTATORS' TRUST IN LARGER MODELS OVER SMALLER ONES

It is common practice to treat the response from a larger model as the chosen (preferred) one and the response from a smaller model as the rejected (dis-preferred) one. Therefore, we obtain new noisy preferences from UFB (each of query has four LLM responses) based on the sizes of models that generate the responses. As shown in Table 9, under this practical noise setting, ROPO still significantly outperforms DPO and other baselines.

### D.3.2 EXPERIMENTS UNDER NOISE COMING FROM LLM PREFERENCE COMPARISONS

We use Llama-3-70B-Instruct [14], which is one of the most advanced open source LLM, to relabel the preferences in UFB. To make the labels as reliable as possible, we instruct the model to list the advantages of each response. The prompt we use is as follows.

```
For the given instruction and two responses (A and B),
please answer:  (1) which response is better overall,
(2) the aspects in which A is superior to B, and (3) the
aspects in which B is superior to A.

Strictly adhere to the following rules:
1.  Answer in bullet points, with each point starting with
a gerund or adjective, excluding the words ``response A''
and ``response B''.
2.  If a response has no superior aspects over another,
output NONE.

Instruction:
{instruction}

Response A
{responseA}

Response B
{responseB}

Your answer MUST STRICTLY follow the format as follows:
**Better**
<Choose A or B>

**Why A is better than B**
```

```
-<First aspect for which A is superior to B>
-<Continue with other points if any>

**Why B is better than A**
-<First aspect for which B is superior to A>
-<Continue with other points if any>
```

However, we observe that **about 30% of the labels are different from those in the original UFB dataset**. This shows that noise are unavoidable due to the diversity in LLM preferences. Then, we train Mistral-7B with different methods on the new noisy dataset. As shown in Table 10, under this practical noise setting, ROPO still significantly outperforms DPO and other baselines.

Table 11: Performance of difference methods on Arena-Hard and MT-Bench. The bold font indicates the best result and an underline indicates the second-best result.

| Benchmark | | Arena-Hard | | | MT-Bench | | |
|---|---|---|---|---|---|---|---|
| Model | Method | 0% | 20% | 40% | 0% | 20% | 40% |
| Mistral-7B | DPO | 10.7 | 8.5 | 6.3 | **7.3** | 5.7 | 4.3 |
| | IPO | 9.2 | 7.9 | 7.3 | 7.2 | 5.9 | 4.9 |
| | rDPO | 9.8 | 9.2 | 8.9 | 7.1 | 6.4 | 5.8 |
| | cDPO | 10.3 | 9.0 | 8.4 | 7.2 | 6.2 | 5.2 |
| | **ROPO** | **13.1** | **12.6** | **11.8** | **7.3** | **6.9** | **6.5** |
| Llama-3-8B | DPO | 17.9 | 15.3 | 14.1 | **7.8** | 6.1 | 4.6 |
| | IPO | 18.6 | 16.8 | 16.0 | 7.4 | 6.3 | 5.0 |
| | rDPO | 18.3 | 17.5 | 17.1 | 7.5 | 6.9 | 6.1 |
| | cDPO | 17.5 | 16.4 | 15.3 | 7.7 | 6.7 | 5.8 |
| | **ROPO** | **20.5** | **19.6** | **18.5** | 7.7 | **7.0** | **6.7** |

## D.4 EXPERIMENTS ON MORE BENCHMARKS

To comprehensively explore the performance of ROPO and baseline methods, we evaluate them on another two widely-used benchmarks, i.e., Arena-Hard [25] and MT-Bench [73]. The details of the benchmarks are as follows.

- MT-Bench [73] contains 80 two-turn conversations, each of which has an open-ended instruction and a corresponding follow-up question. Due to the well-designed questions and the wide coverage of topics, MT-Bench has become a widely-used benchmark to evaluate the multi-turn conversational and instruction-following abilities of AI models.
- Arena-Hard [25] is a challenging benchmark containing 500 single-turn conversations. Compared to AlpacaEval and MT-Bench, Arena-Hard features better model separability, tighter confidence intervals, and achieves a correlation of 98.6% with Chatbot Arena rankings [9].

We evaluate ROPO and baseline methods using Mistral-7B and Llama-3-8B [14]. For Mistral-7B, we use the same models as evaluated on AlpacaEval in the main experiments. For Llama-3-8B, we first train a Llama-3-8B-Base[5] on UltraChat-200k[6] to obtain an SFT model (one epoch with the learning rate of 1e-5, global batch size of 128, weight decay of 0.1, and a consine-type learning rate scheduler), and then continue training with ROPO and baseline methods. The results are shown in Table 11. As observed, under various artificial noise levels, ROPO consistently outperforms baseline methods in most cases and demonstrates superior robustness in noisy scenarios.

## D.5 EXPERIMENTS OF COMBINING DPO WITH NOISY SAMPLES FILTERING AND REJECTION SAMPLING

As shown in Figure 2, the distributions of the DPO loss on clean and noisy samples are very similar, and the difference gradually decreases as the training proceeds. This shows that the DPO loss is

---

[5]https://huggingface.co/meta-llama/Meta-Llama-3-8B
[6]https://huggingface.co/datasets/HuggingFaceH4/ultrachat_200k

Table 12: Win rates (%) of **different variants of DPO vs SFT targets** under 0% and 20% artificial noise, evaluated by GPT-4 on AlpacaEval. The base model is Mistral-7B and the training dataset is UFB.

|  | 0% | 20% |
|---|---|---|
| DPO | **90.60** | **86.21** |
| DPO + NSF | 90.06 | 85.09 |
| DPO + NSF + RS | 90.31 | 84.97 |

Table 13: Human evaluation of ROPO vs DPO and ROPO vs rDPO on AlpacaEval. The base model is Mistral-7B and the training dataset is UFB. The #(Win), #(Tie), and #(Lose) are the numbers of ROPO's wins, ties, and ROPO's losses.

| Artificial Noise Ratio | 0% | | | | 20% | | | |
|---|---|---|---|---|---|---|---|---|
|  | #(Win) | #(Tie) | #(Lose) | WR (%) | #(Win) | #(Tie) | #(Lose) | WR (%) |
| ROPO vs DPO | 77 | 69 | 54 | 55.8 | 103 | 59 | 38 | 66.5 |
| ROPO vs rDPO | 84 | 63 | 53 | 57.8 | 89 | 64 | 47 | 60.5 |

prone to overfitting to noise, hence cannot serve as a reliable measure of model uncertainty in noisy scenarios. In this section, to further support our claim, we conduct experiments of combining DPO with noisy samples filtering (NSF) and rejection sampling (RS) using Mistral-7B as the base model and UFB as the training dataset. Please note that our proposed robustness-guided RS only works on the filtered samples, so we do not conduct experiments combining DPO and RS alone. The results are shown in Table 12. As can be seen, the incorporation of noisy samples filtering and rejection sampling degrades the performance of DPO, especially at 20% artificial noise.

### D.6    HUMAN EVALUATION

We invite four lab members with no conflicts of interest to this paper to serve as volunteers to conduct human evaluations. Two of them are PhDs and the other two are doctoral students, so we believe that they have the ability to understand the evaluation rules and make reliable judgments.

We randomly select 200 queries from the AlpacaEval benchmark. Then, we pair the corresponding responses of ROPO, DPO, and rDPO under 0% and 20% artificial noise to form four groups: (1) ROPO vs DPO under 0% artificial noise, (2) ROPO vs rDPO under 0% artificial noise, (3) ROPO vs DPO under 20% artificial noise, and (4) ROPO vs rDPO under 20% artificial noise.

For each group, we randomly shuffle the order of the queries and the order of responses in each pair. Each volunteer is in charge of one group. None of the volunteers know which method corresponds to each response. They are asked to compare the responses in 200 pairs and choose the better one. If they are unsure about which response is better, they can choose "Tie". During the evaluation process, we allow the volunteers to use translation tools and search engines.

We count the number of ROPO's wins, ties, and losses, and compute the win rate of ROPO by $\Omega = \frac{\#(\text{Win}) + \#(\text{Tie})/2}{200}$. The results are shown in Table 13. We have the following interesting observations from the table: (1) The win rate of ROPO against DPO and rDPO is consistently over 55%, demonstrating ROPO's advantages over the baselines. (2) As the artificial noise rate increases, the win rate of ROPO increases to more than 60%, which shows the superiority of ROPO in noisy scenarios. (3) All four volunteers give at least 29% tie judgments, indicating the limitations of human evaluation: it is challenging for most human evaluators to make reliable evaluations on difficult tasks such as long-context reasoning, coding, mathematics, etc. This highlights the importance of developing automated LLM evaluation tools.

### D.7    EXPERIMENTS OF APPLYING REGULARIZATION STRATEGIES TO DPO

In experiments in the main text, we have evaluate the performance of label smoothing (i.e., cDPO and rDPO) under noisy scenarios. The label smoothing techniques can be seen as regularization

Table 14: Win rates (%) of **DPO with confidence penalty vs SFT targets** under 20% and 40% artificial noise, evaluated by GPT-4 on AlpacaEval. The base model is Mistral-7B and the training dataset is UFB.

|  | 20% | 40% |
|---|---|---|
| DPO | 86.21 | **82.67** |
| DPO + CP | **86.96** | 81.86 |

strategies applied to DPO. As shown in the experiments, they bring performance improvements over DPO under 20% and 40% artificial noise, but underperform ROPO. Their limited effectiveness might be attributed to the fact that rDPO and cDPO are noise-tolerant only under specific conditions: when the hyperparameter $\varepsilon$ exactly matches the noise proportion for rDPO, and when $\varepsilon = 0.5$ for cDPO. Achieving these conditions in practice is challenging due to the lack of prior knowledge about the exact noise proportion.

In this section, we explore another two widely-used types of regularization strategies in noisy scenarios, i.e., the *normalized negative loss* and *confidence penalty*.

- Normalized negative loss (NNL) [68], such as normalized negative cross entropy (NNCE) and normalized negative focal loss (NNFL), are shown to be effective when combined with the cross-entropy loss (i.e., the DPO loss in preference optimization). However, when the problem is binary classification like preference comparison, NNCE and NNFL degenerate into constant terms. Specifically, for a sample $(\mathbf{x}, \mathbf{y}_1, \mathbf{y}_2, \mathbf{y}_1 \succ \mathbf{y}_2 \mid \mathbf{x})$, if we denote $P = \sigma(r(\mathbf{x}, \mathbf{y}_1) - r(\mathbf{x}, \mathbf{y}_2))$, then we have

$$\ell_{\mathrm{nnce}} = 1 - \frac{-\log\min(P, 1-P) + \log P}{-2\log\min(P, 1-P) + \log P + \log(1-P)}$$

$$= \begin{cases} 1, & \text{if } P \le 0.5, \\ 0, & \text{if } P > 0.5, \end{cases}$$

and

$$\ell_{\mathrm{nnfl}} = 1 - \frac{-(1 - \min(P, 1-P))^\gamma \log\min(P, 1-P) + (1-P)^\gamma \log P}{-2(1 - \min(P, 1-P))^\gamma \log\min(P, 1-P) + (1-P)^\gamma \log P + P^\gamma \log(1-P)}$$

$$= \begin{cases} 1, & \text{if } P \le 0.5, \\ 0, & \text{if } P > 0.5. \end{cases}$$

Therefore, NNL does not work for DPO.

- Confidence penalty (CP) [40] is an entropy-aware regularizer for the cross-entropy loss, which prevents the model from making overconfident inferences. Specifically, for a sample $(\mathbf{x}, \mathbf{y}_1, \mathbf{y}_2, \mathbf{y}_1 \succ \mathbf{y}_2 \mid \mathbf{x})$, if we denote $P_\theta = \sigma(r_\theta(\mathbf{x}, \mathbf{y}_1) - r_\theta(\mathbf{x}, \mathbf{y}_2))$, CP computes the entropy by

$$H_\theta = -P_\theta \log P_\theta - (1 - P_\theta)\log(1 - P_\theta).$$

Then, the CP regularizer is

$$\ell_{\mathrm{cp}} = -\lambda \max(0, \gamma - H_\theta).$$

We combine DPO with CP and tune the hyperparameters $\lambda$ and $\gamma$ in the range of $\lambda \in \{0.01, 0.1\}$ and $\gamma \in \{0.1, 0.25, 0.5\}$. As shown in Table 14, we do not observe a significant improvement over DPO in noisy scenarios. We speculate that the limited effectiveness of CP is because CP has no guaranteed noise-tolerance.

# E  MATHEMATICAL DERIVATIONS AND THEORETICAL ANALYSIS

## E.1  PROOF OF THEOREM 3.1

*Proof.* As $\sum_{i=1}^{N} w_i = N_\rho$ is a hyperplane and $w_i \in [0, 1]$ for $i = 1, \ldots, N$, $S \triangleq \{\mathbf{w} : w_i \in [0, 1], \sum_{i=1}^{N} w_i = N_\rho\}$ is compact. Because $\Theta$ is compact, $\Theta \times S$ is compact. Therefore, the continuous $\frac{1}{N}\sum_{i=1}^{N} w_i \ell\left(\theta; \mathbf{x}^{(i)}, \mathbf{y}_1^{(i)}, \mathbf{y}_2^{(i)}, \hat{c}^{(i)}, \pi_\theta\right)$ admits an optimal solution $(\theta^*, \mathbf{w}^*)$ on $\Theta \times S$.

Assume that $\ell\left(\theta^*; \mathbf{x}^{(i_1)}, \mathbf{y}_1^{(i_1)}, \mathbf{y}_2^{(i_1)}, \pi_{\theta^*}\right) < \cdots < \ell\left(\theta^*; \mathbf{x}^{(i_N)}, \mathbf{y}_1^{(i_N)}, \mathbf{y}_2^{(i_N)}, \pi_{\theta^*}\right)$ but with $w_{i_j}^* <$ 1 for some $1 \leq j \leq N_\rho$. Then, we have

$$\sum_{k=1}^{N_\rho} w_{i_k}^* < 1 + (N_\rho - 1) = N_\rho, \tag{17}$$

hence there exists $w_{i_l}^* > 0$ for some $N_\rho < l \leq N$. By letting $w_{i_j}' = 1$, $w_{i_l}' = w_{i_j}^* + w_{i_l}^* - 1$, and $w_{i_k}' = w_{i_k}^*$ for $k \neq j, l$, we have $\sum_{k=1}^N w_{i_k}' = 1$ and

$$
\begin{aligned}
\frac{1}{N}\sum_{i=1}^N w_i' \ell\left(\theta^*; \mathbf{x}^{(i)}, \mathbf{y}_1^{(i)}, \mathbf{y}_2^{(i)}, \hat{c}^{(i)}, \pi_{\theta^*}\right) &= \frac{1}{N}\sum_{k \neq j,l} w_{i_k}' \ell\left(\theta^*; \mathbf{x}^{(i_k)}, \mathbf{y}_1^{(i_k)}, \mathbf{y}_2^{(i_k)}, \hat{c}^{(i_k)}, \pi_{\theta^*}\right) \\
&\quad + w_{i_j}' \ell\left(\theta^*; \mathbf{x}^{(i_j)}, \mathbf{y}_1^{(i_j)}, \mathbf{y}_2^{(i_j)}, \hat{c}^{(i_j)}, \pi_{\theta^*}\right) \\
&\quad + w_{i_l}' \ell\left(\theta^*; \mathbf{x}^{(i_l)}, \mathbf{y}_1^{(i_l)}, \mathbf{y}_2^{(i_l)}, \hat{c}^{(i_l)}, \pi_{\theta^*}\right) \\
&< \frac{1}{N}\sum_{k \neq j,l} w_{i_k}^* \ell\left(\theta^*; \mathbf{x}^{(i_k)}, \mathbf{y}_1^{(i_k)}, \mathbf{y}_2^{(i_k)}, \hat{c}^{(i_k)}, \pi_{\theta^*}\right) \\
&\quad + w_{i_j}^* \ell\left(\theta^*; \mathbf{x}^{(i_j)}, \mathbf{y}_1^{(i_j)}, \mathbf{y}_2^{(i_j)}, \hat{c}^{(i_j)}, \pi_{\theta^*}\right) \\
&\quad + w_{i_l}^* \ell\left(\theta^*; \mathbf{x}^{(i_l)}, \mathbf{y}_1^{(i_l)}, \mathbf{y}_2^{(i_l)}, \hat{c}^{(i_l)}, \pi_{\theta^*}\right) \\
&= \frac{1}{N}\sum_{i=1}^N w_i^* \ell\left(\theta^*; \mathbf{x}^{(i)}, \mathbf{y}_1^{(i)}, \mathbf{y}_2^{(i)}, \hat{c}^{(i)}, \pi_{\theta^*}\right), \tag{18}
\end{aligned}
$$

which leads to a contradiction. Therefore, we must have $w_{i_k}^* = 1$ for $1 \leq k \leq N_\rho$ and $w_{i_k}^* = 0$ for $N_\rho < k \leq N$. $\qquad\square$

### E.2 Proof of Theorem 3.2

*Proof.* For $\ell = \ell_{\mathrm{dpo}}$, we have

$$
\begin{aligned}
&\mathbb{E}_{(\mathbf{x}, \mathbf{y}_1, \mathbf{y}_2, \hat{c}) \sim \mathcal{D}_\eta}[\ell(\theta; \mathbf{x}, \mathbf{y}_1, \mathbf{y}_2, \hat{c}, \pi_\theta)] \\
&= \mathbb{E}_{(\mathbf{x}, \mathbf{y}_1, \mathbf{y}_2)}\mathbb{E}_{c|\mathbf{x}, \mathbf{y}_1, \mathbf{y}_2}\mathbb{E}_{\hat{c}|\mathbf{x}, \mathbf{y}_1, \mathbf{y}_2, c}[\ell(\theta; \mathbf{x}, \mathbf{y}_1, \mathbf{y}_2, \hat{c}, \pi_\theta)] \\
&= \mathbb{E}_{(\mathbf{x}, \mathbf{y}_1, \mathbf{y}_2)}\Big[ (P^*(\mathbf{y}_1 \succ \mathbf{y}_2 \mid \mathbf{x})(1 - \eta) + (1 - P^*(\mathbf{y}_1 \succ \mathbf{y}_2 \mid \mathbf{x}))\eta) \cdot \ell(\theta; \mathbf{x}, \mathbf{y}_1, \mathbf{y}_2, 0, \pi_\theta) \\
&\qquad\qquad + (P^*(\mathbf{y}_1 \succ \mathbf{y}_2 \mid \mathbf{x})\eta + (1 - P^*(\mathbf{y}_1 \succ \mathbf{y}_2 \mid \mathbf{x}))(1 - \eta)) \cdot \ell(\theta; \mathbf{x}, \mathbf{y}_1, \mathbf{y}_2, 1, \pi_\theta)\Big] \\
&= \mathbb{E}_{(\mathbf{x}, \mathbf{y}_1, \mathbf{y}_2)}\Big[ - (P^*(\mathbf{y}_1 \succ \mathbf{y}_2 \mid \mathbf{x}) + \eta - 2P^*(\mathbf{y}_1 \succ \mathbf{y}_2 \mid \mathbf{x})\eta) \log P_\theta(\mathbf{y}_1 \succ \mathbf{y}_2 \mid \mathbf{x}) \\
&\qquad\qquad - (2P^*(\mathbf{y}_1 \succ \mathbf{y}_2 \mid \mathbf{x})\eta + 1 - P^*(\mathbf{y}_1 \succ \mathbf{y}_2 \mid \mathbf{x}) - \eta) \log(1 - P_\theta(\mathbf{y}_1 \succ \mathbf{y}_2 \mid \mathbf{x}))\Big].
\end{aligned}
\tag{19}
$$

Consider

$$f(p) = -(p^* + \eta - 2p^*\eta) \log p - (2p^*\eta + 1 - p^* - \eta) \log(1 - p), \tag{20}$$

we have

$$f'(p) = -\frac{p^* + \eta - 2p^*\eta}{p} + \frac{2p^*\eta + 1 - p^* - \eta}{1 - p}. \tag{21}$$

From $f'(p)$ we know that $f$ decrease when $p \leq p^* + \eta - 2p^*\eta$ and increases when $p \geq p^* + \eta - 2p^*\eta$, which means that $f$ reaches its minimum at $p_0 = p^* + (1 - 2p^*)\eta$.

Therefore, Eq. (19) reaches its minimum when

$$P_{\theta_\eta^*}(\mathbf{y}_1 \succ \mathbf{y}_2 \mid \mathbf{x}) = P^*(\mathbf{y}_1 \succ \mathbf{y}_2 \mid \mathbf{x}) + (1 - 2P^*(\mathbf{y}_1 \succ \mathbf{y}_2 \mid \mathbf{x}))\eta \tag{22}$$

for any $(\mathbf{x}, \mathbf{y}_1, \mathbf{y}_2)$. Specifically, for $\eta = 0$, we have $P_{\theta^*}(\mathbf{y}_1 \succ \mathbf{y}_2 \mid \mathbf{x}) = P^*(\mathbf{y}_1 \succ \mathbf{y}_2 \mid \mathbf{x})$, which leas to

$$\left|P_{\theta_\eta^*}(\mathbf{y}_1 \succ \mathbf{y}_2 \mid \mathbf{x}) - P_{\theta^*}(\mathbf{y}_1 \succ \mathbf{y}_2 \mid \mathbf{x})\right| = 2\eta\left|P^*(\mathbf{y}_1 \succ \mathbf{y}_2 \mid \mathbf{x}) - 1/2\right|. \tag{23}$$

$\square$

### E.3 Proof of Theorem 3.3

*Proof.* For samples $(\mathbf{x}^{(1)}, \mathbf{y}_1^{(1)}, \mathbf{y}_2^{(1)}, \hat{c}^{(1)} = c^{(1)})$ and $(\mathbf{x}^{(2)}, \mathbf{y}_1^{(2)}, \mathbf{y}_2^{(2)}, \hat{c}^{(2)} = 1 - c^{(2)})$, according to Eq. (22), we have

$$P_{\theta_\eta^*}\left(\mathbf{x}^{(1)}, \mathbf{y}_1^{(1)}, \mathbf{y}_2^{(1)}, \hat{c}^{(1)}\right) = P_{\theta_\eta^*}\left(\mathbf{x}^{(1)}, \mathbf{y}_1^{(1)}, \mathbf{y}_2^{(1)}, c^{(1)}\right)$$
$$= P^*(c^{(1)}) + (1 - 2P^*(c^{(1)}))\eta \tag{24}$$

and

$$P_{\theta_\eta^*}\left(\mathbf{x}^{(2)}, \mathbf{y}_1^{(2)}, \mathbf{y}_2^{(2)}, \hat{c}^{(2)}\right) = P_{\theta_\eta^*}\left(\mathbf{x}^{(2)}, \mathbf{y}_1^{(2)}, \mathbf{y}_2^{(2)}, 1 - c^{(2)}\right) \tag{25}$$
$$= P^*(1 - c^{(2)}) + (1 - 2P^*(1 - c^{(2)}))\eta$$
$$= 1 - P^*(c^{(2)}) + (2P^*(c^{(2)}) - 1)\eta. \tag{26}$$

Therefore, to ensure that

$$\ell_{\text{dpo}}\left(\mathbf{x}^{(1)}, \mathbf{y}_1^{(1)}, \mathbf{y}_2^{(1)}, \hat{c}^{(1)}\right) - \ell_{\text{dpo}}\left(\mathbf{x}^{(2)}, \mathbf{y}_1^{(2)}, \mathbf{y}_2^{(2)}, \hat{c}^{(2)}\right) < 0, \tag{27}$$

we must have

$$-\log\left(P^*(c^{(1)}) + (1 - 2P^*(c^{(1)}))\eta - \varepsilon\right) < -\log\left(1 - P^*(c^{(2)}) + (2P^*(c^{(2)}) - 1)\eta + \varepsilon\right), \tag{28}$$

which is equivalent to

$$\varepsilon < \frac{1 - 2\eta}{2}\left(P^*(c^{(1)}) + P^*(c^{(2)}) - 1\right). \tag{29}$$

$\square$

### E.4 Detailed Derivation of Eq. (9)

From the definition of $w_{\text{ropo}}$ we have

$$w_{\text{ropo}} = \frac{4\alpha}{(1+\alpha)^2}\sigma(\Delta(\mathbf{y}_2, \mathbf{y}_1, \mathbf{x})) + \frac{4\alpha^2}{(1+\alpha)^2}\sigma(\Delta(\mathbf{y}_2, \mathbf{y}_1, \mathbf{x}))\sigma(\Delta(\mathbf{y}_1, \mathbf{y}_2, \mathbf{x})). \tag{30}$$

According to Eq. (7) we know that

$$-\int \beta \frac{4\alpha}{(1+\alpha)^2}\sigma(\Delta(\mathbf{y}_2, \mathbf{y}_1, \mathbf{x}))\nabla \log \frac{\pi_\theta(\mathbf{y}_1 \mid \mathbf{x})}{\pi_\theta(\mathbf{y}_2 \mid \mathbf{x})} \, d\theta = \frac{4\alpha}{(1+\alpha)^2}\ell_{\text{dpo}}. \tag{31}$$

Beside, note that for $\sigma(x) = \frac{e^x}{1+e^x}$, we have

$$\sigma'(x) = \left(\frac{e^x}{1+e^x}\right)' = \frac{e^x(1+e^x) - e^x \cdot e^x}{(1+e^x)^2} = \frac{e^x}{(1+e^x)^2} = \frac{e^x}{1+e^x} \cdot \frac{1}{1+e^x} = \sigma(x)\sigma(-x) \tag{32}$$

and

$$\sigma'(-x) = -\sigma(x)\sigma(-x). \tag{33}$$

Letting

$$t(\theta) = \beta \log \frac{\pi_\theta(\mathbf{y}_1 \mid \mathbf{x})}{\pi_{\mathrm{ref}}(\mathbf{y}_1 \mid \mathbf{x})} - \beta \log \frac{\pi_\theta(\mathbf{y}_2 \mid \mathbf{x})}{\pi_{\mathrm{ref}}(\mathbf{y}_2 \mid \mathbf{x})}, \tag{34}$$

we have

$$\nabla_\theta t(\theta) = \beta \nabla \log \frac{\pi_\theta(\mathbf{y}_1 \mid \mathbf{x})}{\pi_\theta(\mathbf{y}_2 \mid \mathbf{x})} \tag{35}$$

Hence,

$$-\frac{4\alpha^2}{(1+\alpha)^2} \int \beta \sigma(\Delta(\mathbf{y}_2, \mathbf{y}_1, \mathbf{x})) \sigma(\Delta(\mathbf{y}_1, \mathbf{y}_2, \mathbf{x})) \nabla \log \frac{\pi_\theta(\mathbf{y}_1 \mid \mathbf{x})}{\pi_\theta(\mathbf{y}_2 \mid \mathbf{x})} \, \mathrm{d}\theta$$

$$= \frac{4\alpha^2}{(1+\alpha)^2} \int \left( -\sigma(t(\theta))\sigma(-t(\theta)) \right) \cdot \left( \beta \nabla \log \frac{\pi_\theta(\mathbf{y}_1 \mid \mathbf{x})}{\pi_\theta(\mathbf{y}_2 \mid \mathbf{x})} \right) \mathrm{d}\theta$$

$$= \frac{4\alpha^2}{(1+\alpha)^2} \int \nabla_{t(\theta)} \sigma(-t(\theta)) \cdot \nabla_\theta t(\theta) \, \mathrm{d}\theta$$

$$= \frac{4\alpha^2}{(1+\alpha)^2} \int \nabla_\theta \sigma(-t(\theta)) \, \mathrm{d}\theta$$

$$= \frac{4\alpha^2}{(1+\alpha)^2} \sigma(-t(\theta))$$

$$= \frac{4\alpha^2}{(1+\alpha)^2} \cdot \sigma \left( \beta \log \frac{\pi_\theta(\mathbf{y}_2 \mid \mathbf{x})}{\pi_{\mathrm{ref}}(\mathbf{y}_2 \mid \mathbf{x})} - \beta \log \frac{\pi_\theta(\mathbf{y}_1 \mid \mathbf{x})}{\pi_{\mathrm{ref}}(\mathbf{y}_1 \mid \mathbf{x})} \right), \tag{36}$$

where we omit the constant term of the primitive function.

### E.5 Proof of Theorem 3.4

*Proof.* For $\ell = \ell_{\mathrm{na}}$, we have

$$\mathbb{E}_{(\mathbf{x}, \mathbf{y}_1, \mathbf{y}_2, \hat{c}) \sim \mathcal{D}_\eta} [\ell(\theta; \mathbf{x}, \mathbf{y}_1, \mathbf{y}_2, \hat{c}, \pi_\theta)]$$

$$= \mathbb{E}_{(\mathbf{x}, \mathbf{y}_1, \mathbf{y}_2)} \mathbb{E}_{c \mid \mathbf{x}, \mathbf{y}_1, \mathbf{y}_2} \mathbb{E}_{\hat{c} \mid \mathbf{x}, \mathbf{y}_1, \mathbf{y}_2, c} [\ell(\theta; \mathbf{x}, \mathbf{y}_1, \mathbf{y}_2, \hat{c}, \pi_\theta)]$$

$$= \mathbb{E}_{(\mathbf{x}, \mathbf{y}_1, \mathbf{y}_2)} \Bigg[ \left( P^*(\mathbf{y}_1 \succ \mathbf{y}_2 \mid \mathbf{x})(1 - \eta) + (1 - P^*(\mathbf{y}_1 \succ \mathbf{y}_2 \mid \mathbf{x}))\eta \right) \cdot \ell(\theta; \mathbf{x}, \mathbf{y}_1, \mathbf{y}_2, 0, \pi_\theta)$$

$$+ \left( P^*(\mathbf{y}_1 \succ \mathbf{y}_2 \mid \mathbf{x})\eta + (1 - P^*(\mathbf{y}_1 \succ \mathbf{y}_2 \mid \mathbf{x}))(1 - \eta) \right) \cdot \ell(\theta; \mathbf{x}, \mathbf{y}_1, \mathbf{y}_2, 1, \pi_\theta) \Bigg]$$

$$= \mathbb{E}_{(\mathbf{x}, \mathbf{y}_1, \mathbf{y}_2)} \Bigg[ \left( P^*(\mathbf{y}_1 \succ \mathbf{y}_2 \mid \mathbf{x}) + \eta - 2P^*(\mathbf{y}_1 \succ \mathbf{y}_2 \mid \mathbf{x})\eta \right) (1 - P_\theta(\mathbf{y}_1 \succ \mathbf{y}_2 \mid \mathbf{x}))$$

$$+ \left( 2P^*(\mathbf{y}_1 \succ \mathbf{y}_2 \mid \mathbf{x})\eta + 1 - P^*(\mathbf{y}_1 \succ \mathbf{y}_2 \mid \mathbf{x}) - \eta \right) P_\theta(\mathbf{y}_1 \succ \mathbf{y}_2 \mid \mathbf{x}) \Bigg]. \tag{37}$$

Consider

$$\begin{aligned} f(p) &= (p^* + \eta - 2p^*\eta)(1 - p) + (2p^*\eta + 1 - p^* - \eta)p \\ &= (1 - 2\eta)(1 - 2p^*)p + (p^* + \eta - 2p^*\eta). \end{aligned} \tag{38}$$

Therefore, when $p^* > 1/2$, $f(p)$ reaches its minimum at $p = 1$; when $p^* < 1/2$, $f(p)$ reaches its minimum at $p = 0$. This means that the optimal point of $f(p)$ is $p_0 = \mathbb{I}(p^* > 1/2)$.

Therefore, Eq. (37) reaches its minimum when

$$P_{\theta_\eta^*}(\mathbf{y}_1 \succ \mathbf{y}_2 \mid \mathbf{x}) = \mathbb{I}\left( P^*(\mathbf{y}_1 \succ \mathbf{y}_2 \mid \mathbf{x}) > \frac{1}{2} \right) \tag{39}$$

for any $(\mathbf{x}, \mathbf{y}_1, \mathbf{y}_2)$. Obviously, we have

$$P_{\theta_\eta^*}(\mathbf{y}_1 \succ \mathbf{y}_2 \mid \mathbf{x}) = P_{\theta^*}(\mathbf{y}_1 \succ \mathbf{y}_2 \mid \mathbf{x}). \tag{40}$$

$\square$

### E.6 PROOF OF THEOREM 3.5

*Proof.* For samples $(\mathbf{x}^{(1)}, \mathbf{y}_1^{(1)}, \mathbf{y}_2^{(1)}, \hat{c}^{(1)} = c^{(1)})$ and $(\mathbf{x}^{(2)}, \mathbf{y}_1^{(2)}, \mathbf{y}_2^{(2)}, \hat{c}^{(2)} = 1 - c^{(2)})$. Without loss of generality, we only need to consider two cases: (1) $c^{(1)} = c^{(2)} = 0$ and (2) $c^{(1)} = 0, c^{(2)} = 1$. For the first case, we have

$$\ell_{\text{na}}\left(\mathbf{x}^{(1)}, \mathbf{y}_1^{(1)}, \mathbf{y}_2^{(1)}, \hat{c}^{(1)}\right) = P_\theta(\mathbf{y}_2^{(1)} \succ \mathbf{y}_1^{(1)} \mid \mathbf{x}) \in [0, \varepsilon) \tag{41}$$

and

$$\ell_{\text{na}}\left(\mathbf{x}^{(2)}, \mathbf{y}_1^{(2)}, \mathbf{y}_2^{(2)}, \hat{c}^{(2)}\right) = P_\theta(\mathbf{y}_1^{(2)} \succ \mathbf{y}_2^{(2)} \mid \mathbf{x}) \in (1 - \varepsilon, 1]. \tag{42}$$

For the second case, we have

$$\ell_{\text{na}}\left(\mathbf{x}^{(1)}, \mathbf{y}_1^{(1)}, \mathbf{y}_2^{(1)}, \hat{c}^{(1)}\right) = P_\theta(\mathbf{y}_2^{(1)} \succ \mathbf{y}_1^{(1)} \mid \mathbf{x}) \in [0, \varepsilon) \tag{43}$$

and

$$\ell_{\text{na}}\left(\mathbf{x}^{(2)}, \mathbf{y}_1^{(2)}, \mathbf{y}_2^{(2)}, \hat{c}^{(2)}\right) = P_\theta(\mathbf{y}_2^{(2)} \succ \mathbf{y}_1^{(2)} \mid \mathbf{x}) \in (1 - \varepsilon, 1]. \tag{44}$$

Therefore, to ensure that

$$\ell_{\text{na}}\left(\mathbf{x}^{(1)}, \mathbf{y}_1^{(1)}, \mathbf{y}_2^{(1)}, \hat{c}^{(1)}\right) < \ell_{\text{na}}\left(\mathbf{x}^{(2)}, \mathbf{y}_1^{(2)}, \mathbf{y}_2^{(2)}, \hat{c}^{(2)}\right), \tag{45}$$

we must have $\varepsilon < \frac{1}{2}$. □

### E.7 rDPO AND cDPO ARE NOT NOISE-TOLERANT IN MOST CASES

*Proof.* According to Lemma 3.2 in [10], the noise-tolerance of rDPO is only guaranteed when the proportion of noise, i.e., $\eta_0$, exactly equals the hyperparameter $\varepsilon$.

Next we show that $\ell_{\text{cdpo}}$ is not noise-tolerant for $\varepsilon \in (0, \frac{1}{2})$. Let

$$\mathcal{L}_{\text{cdpo}}(\theta) = \mathbb{E}_{(\mathbf{x}, \mathbf{y}_1, \mathbf{y}_2, c) \sim \mathcal{D}}[\ell_{\text{cdpo}}(\theta; \mathbf{x}, \mathbf{y}_1, \mathbf{y}_2, c, \pi_\theta)],$$
$$\mathcal{L}_{\text{cdpo}}^{\eta_0}(\theta) = \mathbb{E}_{(\mathbf{x}, \mathbf{y}_1, \mathbf{y}_2, \hat{c}) \sim \mathcal{D}_{\eta_0}}[\ell_{\text{cdpo}}(\theta; \mathbf{x}, \mathbf{y}_1, \mathbf{y}_2, \hat{c}, \pi_\theta)],$$

and assume that $\theta^*$ and $\theta_{\eta_0}^*$ are the minimizers of $\mathcal{L}_{\text{cdpo}}$ and $\mathcal{L}_{\text{cdpo}}^{\eta_0}$, respectively. For any $\theta$ in the space of parameters, we have

$$\mathcal{L}_{\text{cdpo}}^{\eta_0}(\theta)$$
$$= \mathbb{E}_{(\mathbf{x}, \mathbf{y}_1, \mathbf{y}_2, c) \sim \mathcal{D}} \mathbb{E}_{\hat{c} \mid (\mathbf{x}, \mathbf{y}_1, \mathbf{y}_2, c)}[\ell_{\text{cdpo}}(\theta; \mathbf{x}, \mathbf{y}_1, \mathbf{y}_2, \hat{c}, \pi_\theta)]$$
$$= \mathbb{E}_{(\mathbf{x}, \mathbf{y}_1, \mathbf{y}_2, c) \sim \mathcal{D}}[(1 - \eta_0)\ell_{\text{cdpo}}(\theta; \mathbf{x}, \mathbf{y}_1, \mathbf{y}_2, c, \pi_\theta) + \eta_0 \ell_{\text{cdpo}}(\theta; \mathbf{x}, \mathbf{y}_1, \mathbf{y}_2, 1 - c, \pi_\theta)]$$
$$= (1 - \eta_0)\mathcal{L}_{\text{cdpo}}(\theta) + \eta_0 \mathbb{E}_{(\mathbf{x}, \mathbf{y}_1, \mathbf{y}_2, c) \sim \mathcal{D}}[\ell_{\text{cdpo}}(\theta; \mathbf{x}, \mathbf{y}_1, \mathbf{y}_2, 1 - c, \pi_\theta)]. \tag{46}$$

Next, we give a counter-example to show that $\ell_{\text{cdpo}}$ is not noise-tolerant. Suppose that

$$P\left((\mathbf{x}, \mathbf{y}_1, \mathbf{y}_2) = (\mathbf{x}^{(0)}, \mathbf{y}_1^{(0)}, \mathbf{y}_2^{(0)})\right) = 1 \quad \text{and} \quad \mathbf{y}_1^{(0)} \succ \mathbf{y}_2^{(0)} \mid \mathbf{x}^{(0)}, \tag{47}$$

where $\mathbf{x}^{(0)}$ is a fixed input and $(\mathbf{y}_1^{(0)}, \mathbf{y}_2^{(0)})$ is a fixed pair of responses. Hence Eq. (46) becomes

$$\mathcal{L}_{\text{cdpo}}^{\eta_0}(\theta)$$
$$= (2\varepsilon\eta_0 - \eta_0 - \varepsilon) \log \sigma \left(\beta \log \frac{\pi_\theta(\mathbf{y}_1^{(0)} \mid \mathbf{x}^{(0)})}{\pi_{\text{ref}}(\mathbf{y}_1^{(0)} \mid \mathbf{x}^{(0)})} - \beta \log \frac{\pi_\theta(\mathbf{y}_2^{(0)} \mid \mathbf{x}^{(0)})}{\pi_{\text{ref}}(\mathbf{y}_2^{(0)} \mid \mathbf{x}^{(0)})}\right)$$
$$+ (\eta_0 + \varepsilon - 2\varepsilon\eta_0 - 1) \log \sigma \left(\beta \log \frac{\pi_\theta(\mathbf{y}_2^{(0)} \mid \mathbf{x}^{(0)})}{\pi_{\text{ref}}(\mathbf{y}_2^{(0)} \mid \mathbf{x}^{(0)})} - \beta \log \frac{\pi_\theta(\mathbf{y}_1^{(0)} \mid \mathbf{x}^{(0)})}{\pi_{\text{ref}}(\mathbf{y}_1^{(0)} \mid \mathbf{x}^{(0)})}\right). \tag{48}$$

Let

$$\Delta(\theta) = \beta \log \frac{\pi_\theta(\mathbf{y}_1^{(0)} \mid \mathbf{x}^{(0)})}{\pi_{\text{ref}}(\mathbf{y}_1^{(0)} \mid \mathbf{x}^{(0)})} - \beta \log \frac{\pi_\theta(\mathbf{y}_2^{(0)} \mid \mathbf{x}^{(0)})}{\pi_{\text{ref}}(\mathbf{y}_2^{(0)} \mid \mathbf{x}^{(0)})}, \tag{49}$$

then Eq. (48) becomes

$$\mathcal{L}_{\text{cdpo}}^{\eta_0}(\theta) = (2\varepsilon\eta_0 - \eta_0 - \varepsilon)\log\sigma(\Delta(\theta)) + (\eta_0 + \varepsilon - 2\varepsilon\eta_0 - 1)\log\sigma(-\Delta(\theta)). \qquad (50)$$

We have

$$\theta^* = \underset{\theta\in\Theta}{\arg\min}\ \mathcal{L}_{\text{cdpo}}$$

$$= \underset{\theta\in\Theta}{\arg\min}\ -\varepsilon\log\sigma(\Delta(\theta)) - (1-\varepsilon)\log\sigma(\Delta(-\theta))$$

$$\in \left\{\theta\in\Theta : \Delta(\theta) = \log\frac{\varepsilon}{1-\varepsilon}\right\}, \qquad (51)$$

and

$$\theta_{\eta_0}^* = \underset{\theta\in\Theta}{\arg\min}\ \mathcal{L}_{\text{cdpo}}^{\eta_0}$$

$$= \underset{\theta\in\Theta}{\arg\min}\ (2\varepsilon\eta_0 - \eta_0 - \varepsilon)\log\sigma(\Delta(\theta)) + (\eta_0 + \varepsilon - 2\varepsilon\eta_0 - 1)\log\sigma(-\Delta(\theta))$$

$$\in \left\{\theta\in\Theta : \Delta(\theta) = \log\frac{\eta_0 + \varepsilon - 2\varepsilon\eta_0}{1 - \eta_0 - \varepsilon + 2\varepsilon\eta_0}\right\}. \qquad (52)$$

Hence $\theta^* = \theta_{\eta_0}^*$ if and only if

$$\frac{\varepsilon}{1-\varepsilon} = \frac{\eta_0 + \varepsilon - 2\varepsilon\eta_0}{1 - \eta_0 - \varepsilon + 2\varepsilon\eta_0}, \qquad (53)$$

which means that $\varepsilon = \frac{1}{2}$. However, $\varepsilon \in (0, \frac{1}{2})$. Therefore, $\theta^* \neq \theta_{\eta_0}^*$ and thus $\ell_{\text{cdpo}}$ is not noise-tolerant. $\qquad\square$

### E.8 IPO IS NOT NOISE-TOLERANT

*Proof.* Let

$$\mathcal{L}_{\text{ipo}}(\theta) = \mathbb{E}_{(\mathbf{x},\mathbf{y}_1,\mathbf{y}_2,c)\sim\mathcal{D}}[\ell_{\text{ipo}}(\theta;\mathbf{x},\mathbf{y}_1,\mathbf{y}_2,c,\pi_\theta)],$$

$$\mathcal{L}_{\text{ipo}}^{\eta_0}(\theta) = \mathbb{E}_{(\mathbf{x},\mathbf{y}_1,\mathbf{y}_2,\hat{c})\sim\mathcal{D}_{\eta_0}}[\ell_{\text{ipo}}(\theta;\mathbf{x},\mathbf{y}_1,\mathbf{y}_2,\hat{c},\pi_\theta)],$$

and assume that $\theta^*$ and $\theta_{\eta_0}^*$ are the minimizers of $\mathcal{L}_{\text{ipo}}$ and $\mathcal{L}_{\text{ipo}}^{\eta_0}$, respectively. For any $\theta$ in the space of parameters, we have

$$\mathcal{L}_{\text{ipo}}^{\eta_0}(\theta)$$

$$= \mathbb{E}_{(\mathbf{x},\mathbf{y}_1,\mathbf{y}_2,c)\sim\mathcal{D}}\mathbb{E}_{\hat{c}|(\mathbf{x},\mathbf{y}_1,\mathbf{y}_2,c)}[\ell_{\text{ipo}}(\theta;\mathbf{x},\mathbf{y}_1,\mathbf{y}_2,\hat{c},\pi_\theta)]$$

$$= \mathbb{E}_{(\mathbf{x},\mathbf{y}_1,\mathbf{y}_2,c)\sim\mathcal{D}}[(1-\eta_0)\ell_{\text{ipo}}(\theta;\mathbf{x},\mathbf{y}_1,\mathbf{y}_2,c,\pi_\theta) + \eta_0\ell_{\text{ipo}}(\theta;\mathbf{x},\mathbf{y}_1,\mathbf{y}_2,1-c,\pi_\theta)]$$

$$= (1-\eta_0)\mathcal{L}_{\text{ipo}}(\theta) + \eta_0\mathbb{E}_{(\mathbf{x},\mathbf{y}_1,\mathbf{y}_2,c)\sim\mathcal{D}}[\ell_{\text{ipo}}(\theta;\mathbf{x},\mathbf{y}_1,\mathbf{y}_2,1-c,\pi_\theta)]. \qquad (54)$$

Next, we give a counter-example to show that $\ell_{\text{ipo}}$ is not noise-tolerant. Suppose that

$$P\left((\mathbf{x},\mathbf{y}_1,\mathbf{y}_2) = (\mathbf{x}^{(0)},\mathbf{y}_1^{(0)},\mathbf{y}_2^{(0)})\right) = 1 \quad\text{and}\quad \mathbf{y}_1^{(0)} \succ \mathbf{y}_2^{(0)} \mid \mathbf{x}^{(0)}, \qquad (55)$$

where $\mathbf{x}^{(0)}$ is a fixed input and $(\mathbf{y}_1^{(0)},\mathbf{y}_2^{(0)})$ is a fixed pair of responses. Hence Eq. (54) becomes

$$\mathcal{L}_{\text{ipo}}^{\eta_0}(\theta)$$

$$= (1-\eta_0)\left(\log\frac{\pi_\theta(\mathbf{y}_1^{(0)}\mid\mathbf{x}^{(0)})}{\pi_{\text{ref}}(\mathbf{y}_1^{(0)}\mid\mathbf{x}^{(0)})} - \log\frac{\pi_\theta(\mathbf{y}_2^{(0)}\mid\mathbf{x}^{(0)})}{\pi_{\text{ref}}(\mathbf{y}_2^{(0)}\mid\mathbf{x}^{(0)})} - \frac{1}{2\beta}\right)^2$$

$$+ \eta_0\left(\log\frac{\pi_\theta(\mathbf{y}_2^{(0)}\mid\mathbf{x}^{(0)})}{\pi_{\text{ref}}(\mathbf{y}_2^{(0)}\mid\mathbf{x}^{(0)})} - \log\frac{\pi_\theta(\mathbf{y}_1^{(0)}\mid\mathbf{x}^{(0)})}{\pi_{\text{ref}}(\mathbf{y}_1^{(0)}\mid\mathbf{x}^{(0)})} - \frac{1}{2\beta}\right)^2. \qquad (56)$$

Let

$$\Delta(\theta) = \log\frac{\pi_\theta(\mathbf{y}_1^{(0)}\mid\mathbf{x}^{(0)})}{\pi_{\text{ref}}(\mathbf{y}_1^{(0)}\mid\mathbf{x}^{(0)})} - \log\frac{\pi_\theta(\mathbf{y}_2^{(0)}\mid\mathbf{x}^{(0)})}{\pi_{\text{ref}}(\mathbf{y}_2^{(0)}\mid\mathbf{x}^{(0)})}, \qquad (57)$$

then Eq. (56) becomes

$$\mathcal{L}_{\text{ipo}}^{\eta_0}(\theta) = (1 - \eta_0)\left(\Delta(\theta) - \frac{1}{2\beta}\right)^2 + \eta_0\left(-\Delta(\theta) - \frac{1}{2\beta}\right)^2$$

$$= (\Delta(\theta))^2 + \frac{2\eta_0 - 1}{\beta}\Delta(\theta) + \frac{1}{4\beta^2}, \tag{58}$$

which is a quadratic function. Hence

$$\theta_{\eta_0}^* \in \left\{\theta \in \Theta : \Delta(\theta) = \frac{1}{2\beta} - \frac{\eta_0}{\beta}\right\}. \tag{59}$$

However,

$$\theta^* = \arg\min_{\theta \in \Theta}\ \mathcal{L}_{\text{ipo}}$$

$$= \arg\min_{\theta \in \Theta}\left(\Delta(\theta) - \frac{1}{2\beta}\right)^2$$

$$\in \left\{\theta \in \Theta : \Delta(\theta) = \frac{1}{2\beta}\right\}, \tag{60}$$

which means that $\theta^* \neq \theta_{\eta_0}^*$. Therefore, $\ell_{\text{ipo}}$ is not noise-tolerant. $\qquad\square$

### E.9 The normalization of $w_{\text{ropo}}$

In Eq. (8), we use $\frac{4\alpha}{(1+\alpha)^2}$ to scale the maximum value of $w_{\text{ropo}}$ to 1. Here, we provide the details about it. Let

$$g(t) = \sigma(t)(1 + \alpha\sigma(-t)) = \frac{e^{2t} + (1+\alpha)e^t}{(1+e^t)^2},$$

where $\alpha > 2$, then we have

$$g'(t) = \frac{(2e^{2t} + (\alpha+2)e^t)(e^{2t} + 2e^t + 1) - (2e^{2t} + 2e^t)(e^{2t} + (\alpha+1)e^t)}{(1+e^t)^4}$$

$$= \frac{1}{e^t(1+e^t)^4} \cdot \left((2-\alpha)e^{2t} + 4e^t + (\alpha+2)\right)$$

$$= \frac{1}{e^t(1+e^t)^4} \cdot (1+e^t)((2-\alpha)e^t + \alpha + 2)$$

$$= \frac{1}{e^t(1+e^t)^3} \cdot ((2-\alpha)e^t + \alpha + 2).$$

Hence, $g(t)$ increases if and only if $(2-\alpha)e^t + \alpha + 2 \geq 0$.

Since $\alpha > 2$, $g(t)$ increases when $t < \log\frac{\alpha+2}{\alpha-2}$ and decreases when $t > \log\frac{\alpha+2}{\alpha-2}$. Therefore, we have

$$\max_t g(t) = g\left(\log\frac{\alpha+2}{\alpha-2}\right) = \frac{(1+\alpha)^2}{4\alpha}.$$

