# OpenReview forum: "ROPO: Robust Preference Optimization for Large Language Models"
_ICLR.cc/2025/Conference — Submitted to ICLR 2025_

### Official Review · Reviewer_okiK · 2024-11-02

**Soundness:** 3
**Presentation:** 3
**Contribution:** 3
**Rating:** 6
**Confidence:** 4

**Summary:**

This paper examines the unavoidable presence of noise in preference learning and its significant impact on the performance of Large LLMs. Previous research has only slightly reduced the negative effects of noise, which persists during the training phase. Additionally, efforts to filter out noisy samples often lead to increased computational costs. To address these challenges, the paper introduces the ROPO framework, which combines noise tolerance and the filtering of noisy samples. It also incorporates the technique of rejection sampling to further enhance performance. Specifically, the authors derive a loss function through mathematical derivation designed to suppress the gradients of samples with high uncertainty. This approach prevents the model from overfitting to noisy samples while simultaneously identifying them. The effectiveness of the ROPO framework is demonstrated across three datasets in both practical and artificially noisy scenarios.

**Strengths:**

1. The author demonstrated through extensive derivations that methods such as DPO are not noise-tolerant and have difficulty distinguishing between noisy and clean samples. Additionally, the gradient weighting strategy of DPO amplifies the impact of noise. The author derived a loss as a regularizer through a conservative gradient weighting strategy to prevent the model from overfitting to noisy samples and to identify noisy samples.

2. The author not only proved the effectiveness of ROPO on artificial noise but also validated that ROPO can still outperform DPO and other baselines in more practical noisy scenarios.

**Weaknesses:**

1. Although the author presented the framework of ROPO in Figure 1, the paper still lacks an overall description of ROPO, making it difficult to understand how the components of ROPO—noisy sample filtering, rejection sampling stages, and noise tolerance training—are integrated and how the method works iteratively. The author could perhaps add some overall descriptions of the framework.

2. ROPO inevitably introduces too many hyperparameters, such as the trade-off hyperparameter alpha and the sample filtering ratio, which seem to require experimental determination. Along with the hyperparameter beta from DPO, does this make the ROPO algorithm more complex? For example, would different tasks require exploring different combinations of hyperparameters, thereby weakening its practical value?

**Questions:**

1. Could you provide a more detailed overall description of the ROPO framework to clarify how the components (noisy sample filtering, rejection sampling stages, and noise tolerance training) are integrated?

2. Can you include details the iterative process of the ROPO method?

3. Do different tasks require extensive hyperparameter tuning, and if so, how does this affect the practical value of the ROPO method?

---

### Official Review · Reviewer_VETN · 2024-11-04

**Soundness:** 3
**Presentation:** 2
**Contribution:** 3
**Rating:** 6
**Confidence:** 4

**Summary:**

This paper introduces the RObust Preference Optimization (ROPO) framework, a method designed to improve preference alignment in large language models (LLMs) by addressing the challenges posed by noisy preference data. ROPO employs a noise-tolerant loss function and an iterative process that integrates noise filtering during training. Additionally, ROPO includes a robustness-guided rejection sampling technique to retain valuable data while filtering noise. Experiments show that ROPO outperforms existing methods under various noisy conditions, offering a scalable and effective approach to aligning LLMs with human preferences without the need for external models.

**Strengths:**

1. An iterative training approach that optimizes LLM performance while filtering out noisy samples.
2. Experimental results demonstrate improvements over DPO.
3. The use of rejection sampling effectively compensates for information lost during the noise filtering step.

**Weaknesses:**

1. While the paper addresses the impact of noisy data, it lacks a clear definition or characterization of what constitutes noisy data and how it is identified.
2. In the loss function, the primary contribution is the addition of a regularization term, which is not significantly different from the original DPO approach, aside from a scaling coefficient applied to the DPO loss.
3. The selection of $\alpha$ is highly variable, making it difficult to determine an optimal value.

**Questions:**

1. Could you provide a clear definition of noise in the original data and compare the characteristics of noisy data with clean data? Estimating the noise rate in the dataset would add valuable context and make the approach more impactful.
2. Why choose $\frac{4 \alpha}{(1+\alpha)^2}$ to normalize the ROPO loss? Does this yield any specific advantages over other functions?
3. Besides ROPO's regularization terms, could alternative regularization strategies be applied, and how would they impact performance?
4. Could the rejection sampling introduce its own form of bias, especially if it favours certain types of responses?
5. Given ROPO’s iterative nature, what is the computational cost relative to simpler, non-iterative methods, especially for very large LLMs?
6. Does the model’s performance depend on specific types or levels of noise, and how would it handle different real-world noise distributions?

---

> ### Comment · Reviewer_VETN · 2024-11-26
>
> Thank you for your detailed rebuttal. I have reviewed it carefully and will maintain my current score.

---

### Official Review · Reviewer_Co4P · 2024-11-04

**Soundness:** 2
**Presentation:** 3
**Contribution:** 2
**Rating:** 5
**Confidence:** 4

**Summary:**

The paper studies preference alignment under the condition when there are poorly-annotated preference pairs. The authors propose a robust preference optimization (ROPO) framework with two key considerations, (1) a noise-robust loss function that suppresses the gradients of samples that the policy model is uncertain about; (2) A robustness-guided rejection sampling technique designed to balance the filtering of noisy samples with the preservation of important information from queries that might otherwise be discarded.

In the experiments, the authors demonstrate that the policy model aligned with ROPO shows the least drop in performance (win rate against a reference model as judged by GPT-4) with an increasing proportion of injected noise in the training data. The injected noise includes both artificial noise, such as flipping the preference labels of training pairs, and practical noise, where responses from a larger model are blindly assumed to be preferred over those from a smaller model.

**Strengths:**

1. The paper presents a well-motivated study on addressing annotator noise in preference alignment, an issue that is critical for developing reliable policy models.

2. The paper provides a thorough and sensible theoretical analysis of DPO's limitations in discriminating between noisy and clean samples. It also demonstrates how the addition of a regularization loss helps mitigate these issues.

**Weaknesses:**

1. Limited test datasets. Performance evaluation is only conducted on AlpacaEval and the test split of Reddit TL;DR, lack of comprehensive results on multiple instruction-following / alignment benchmarks, such as Wildbench, Arena-Hard, MT-Bench, etc.

2. The paper consider using loss values to identify model-uncertain samples in the robustness-guided rejection sampling procedure as a major contribution. Yet, there has already been several related works, like [1].

[1] Secrets of RLHF in Large Language Models Part II: Reward Modeling.

3. Lack of human evaluation. The analysis is based on GPT-4, which can be biased in its evaluation.

**Questions:**

(1) Only one type of practical noise is considered in the paper, specifically, the assumption that annotators inherently favor outputs from larger models over those from smaller ones. What are other type of practical noises?

(2) The authors mention ROPO is an iterative alignment approach. How the iterative process takes place? It is unclear based on the methodology descriptions in the paper. The authors may provide a detailed algorithm sketch to describe the iterative process.

---

> ### Author Response · Authors · 2024-12-03
> **We remain hopeful and sincerely look forward to receiving your valuable feedback**
>
> Dear Reviewer Co4P,
>
> We are writing as the authors of the paper "ROPO: Robust Preference Optimization for Large Language Models" (ID: 6273). We sincerely appreciate the time and effort you have devoted to reviewing our paper.
>
> While we have not yet received your feedback during the discussion phase, **we fully understand that you may be managing other pressing commitments or navigating a hectic schedule**. We would like to take this opportunity to emphasize two key points below, which we hope will be helpful for further discussions.
>
> - **We humbly believe that you recognize the value and significance of our work**, as reflected in your review, where you kindly describe our work as "*presenting a well-motivated study*", "*addressing an issue that is critical*", and "*providing a thorough and sensible theoretical analysis*".
> - Furthermore, **we humbly believe that our rebuttal has adequately addressed your concerns**. We have carefully revised our paper and provided comprehensive experiments, analyses, and discussions to improve our submission. To assist in **saving your valuable time**, we have also summarized the key points of our rebuttal **for your convenience**.
>
> As the discussion period will conclude in about **2 hours**, we remain hopeful and sincerely look forward to receiving your valuable feedback, should your schedule allow. If our rebuttal has properly addressed your concerns, we would greatly appreciate it if you could raise your score ("*5: marginally below the acceptance threshold*"). If not, please let us know your remaining concerns or questions. We will do our utmost to address your further concerns and answer any questions.
>
> Thank you again for your time, effort, and thoughtful consideration.
>
> Best regards,
>
> Authors of #6273

---

> ### Author Response · Authors · 2024-12-04
> **Thank you once again for your time and efforts.**
>
> Dear Reviewer Co4P,
>
> We are writing as the authors of the paper "ROPO: Robust Preference Optimization for Large Language Models" (ID: 6273).
>
> Before the discussion period ends, we would like to express our sincere gratitude to you once again. We regret not having an in-depth discussion with you and sincerely hope that our rebuttal has properly addressed your concerns.
>
> Thank you for your constructive suggestions and valuable comments.
>
> Best regards,
>
> Authors of #6273

---

### Meta-Review · Area_Chair_WFbC · 2024-12-16

**Metareview:**

The paper presents the RObust Preference Optimization (ROPO) framework, a novel approach to improving preference alignment in LLMs under noisy conditions. By integrating a noise-tolerant loss function and robustness-guided rejection sampling, ROPO aims to mitigate the impact of noisy preference data while preserving valuable information. Experimental results across various datasets demonstrate that ROPO outperforms existing methods in both real and artificially noisy scenarios, achieving better alignment with human preferences without increasing computational overhead.

Strengths:

* The paper addresses the important challenge of annotator noise in preference alignment, which is a critical issue for the development of reliable policy models.
 * It includes a theoretical analysis that outlines DPO's limitations in distinguishing between noisy and clean samples and introduces a regularization loss as a mitigation strategy.
* Experimental results show improvements over DPO in noisy scenarios.

Weaknesses:

* Novelty: The contribution of the proposed approach is limited, as the primary contribution is the addition of a regularization term, which is not significantly different from the original DPO approach. Furthermore, the use of loss values to identify model-uncertain samples in the robustness-guided rejection sampling procedure closely resembles concepts explored in prior work.
 * Complexity: The method introduces multiple hyperparameters, such as the trade-off parameter alpha, the sample filtering ratio, and beta from DPO, which require extensive experimental tuning. This added complexity could reduce the practical value of the approach.
 * Evaluation: The evaluation is constrained by the use of limited test datasets, primarily AlpacaEval and Reddit. This lack of coverage across multiple instruction-following benchmarks such as Wildbench, Arena-Hard, and MT-Bench limits the generalizability of the findings. Additionally, the paper does not sufficiently define or characterize noisy data, making it unclear how noise is identified and mitigated.
* Clarity: While Figure 1 outlines the framework of ROPO, the absence of an overall description makes it challenging to understand how ROPO's components—e.g., noisy sample filtering, rejection sampling, and noise tolerance training—are integrated into an iterative workflow. This could make the work difficult to reproduce.

The evaluation and clarity issues were partially addressed during the discussion period, with the addition of new experiments and a more formal presentation of ROPO in the appendix. However, the addition of significant new experiments (e.g., key results such as human evaluation) would ideally require a fresh round of reviewing. Furthermore, reviewers remain concerned about the novelty and complexity weaknesses mentioned above, even after considering the authors' latest comments and updated paper. In light of these concerns, I recommend rejection.

**Additional Comments On Reviewer Discussion:**

The author-reviewer discussion led to improvements in the evaluation and clarity of the paper. The authors also provided justifications regarding the aforementioned concerns about novelty and complexity (e.g., they argued that the method is relatively robust to hyperparameter choice).

While the evaluation and clarity issues were partially addressed, the reviewer-AC discussion focused on the other two weaknesses. Reviewers still felt that the novelty was not particularly strong, as numerous other works are based on DPO modifications, and similar ideas have been explored before (see reviews/discussions for details). Additionally, the reviewers were not convinced by the authors' rebuttal on complexity, and the next version of the paper may require more analyses to support the authors' claims.

---

### Decision · Program_Chairs · 2025-01-22

Reject